# Complexome profile of *Toxoplasma gondii* mitochondria identifies divergent subunits of respiratory chain complexes including new subunits of cytochrome *bc*$_1$ complex

Andrew E. Maclean[1]*, Hannah R. Bridges[2], Mariana F. Silva[1,3], Shujing Ding[2], Jana Ovciarikova[1], Judy Hirst[2], Lilach Sheiner[1]*

**1** Wellcome Centre for Integrative Parasitology, University of Glasgow, Glasgow, United Kingdom, **2** MRC Mitochondrial Biology Unit, University of Cambridge, Cambridge, United Kingdom, **3** Institute of Biomedical Sciences, Federal University of Uberlândia, Uberlândia, Brazil

\* andrew.maclean@glasgow.ac.uk (AEM); lilach.sheiner@glasgow.ac.uk (LS)

**Data Availability Statement:** All relevant data are within the manuscript and its Supporting Information files. The proteomic complexome data

## Abstract

The mitochondrial electron transport chain (mETC) and F$_1$F$_o$-ATP synthase are of central importance for energy and metabolism in eukaryotic cells. The Apicomplexa, important pathogens of humans causing diseases such as toxoplasmosis and malaria, depend on their mETC in every known stage of their complicated life cycles. Here, using a complexome profiling proteomic approach, we have characterised the *Toxoplasma* mETC complexes and F$_1$F$_o$-ATP synthase. We identified and assigned 60 proteins to complexes II, IV and F$_1$F$_o$-ATP synthase of *Toxoplasma*, of which 16 have not been identified previously. Notably, our complexome profile elucidates the composition of the *Toxoplasma* complex III, the target of clinically used drugs such as atovaquone. We identified two new homologous subunits and two new parasite-specific subunits, one of which is broadly conserved in myzozoans. We demonstrate all four proteins are essential for complex III stability and parasite growth, and show their depletion leads to decreased mitochondrial potential, supporting their assignment as complex III subunits. Our study highlights the divergent subunit composition of the apicomplexan mETC and F$_1$F$_o$-ATP synthase complexes and sets the stage for future structural and drug discovery studies.

## Author summary

Apicomplexan parasites, such as *Toxoplasma* and *Plasmodium*, cause diseases of global importance, such as toxoplasmosis and malaria. The mitochondrial electron transport chain (mETC) and F$_1$F$_o$-ATP synthase, which provide the parasite with energy and important metabolites, are essential for parasite function. Here, using a proteomic technique called complexome profiling, we report the composition of the *Toxoplasma* mETC and F$_1$F$_o$-ATP synthase. In particular, we reveal the compositions of complexes II and III for the first time. Complex III is an important drug target, yet its full protein composition

is deposited in the CEDAR database https://www3.cmbi.umcn.nl/cedar/browse/experiments/CRX27.

**Funding:** This work was funded by: The Biotechnology and Biological Sciences Research Council (BBSRC) (grant BB/N003675/1 to L.S.); a Wellcome Investigator Award (217173/Z/19/Z) to L.S.; the Medical Research Council (grant MC_UU_00015/2 to J.H.); an MVLS Wellcome Institutional Strategic Support Fund (ISSF) ECR Catalyst Grant to A.E.M. and a BBSRC FTMA (grant BB/R506576/1 to A.E.M). L.S. is a Royal Society of Edinburgh Personal Research Fellow. M.F.S has a Coordenação de Aperfeiçoamento de Pessoal de Nível Superior - Brasil (Capes) - Finance Code 001 - studentship. The funders had no role in study design, data collection and analysis, decision to publish, or preparation of the manuscript.

**Competing interests:** The authors have declared that no competing interests exist.

was unknown. We identify new parasite-specific complex III subunits and demonstrate that they are essential for parasite survival and for proper functioning of the mETC. Our study highlights the divergent nature of the apicomplexan mETC and $F_1F_o$-ATP synthase.

## Introduction

The mitochondrial electron transport chain (mETC), a series of protein complexes in the inner mitochondrial membrane, is essential in nearly all eukaryotic cells. Typically, four complexes (complexes I-IV) (**Fig 1A**) transport electrons that are harvested from metabolic pathways, such as fatty acid oxidation and the TCA cycle, to oxygen, the final electron acceptor. Complexes I, III and IV in the mETC couple electron transfer to proton translocation across the inner mitochondrial membrane to form an electrochemical potential, which is then utilised by the mitochondrial $F_1F_o$-ATP synthase (ATP synthase) to generate ATP: this whole process is known as oxidative phosphorylation. In addition to generating ATP, the electrochemical potential produced by the mETC powers many other essential functions, including the import of mitochondrial proteins [1] and the synthesis of pyrimidines [2].

Apicomplexans are eukaryotic unicellular parasites which cause diseases of global importance in humans and livestock. Malaria, caused by *Plasmodium* spp, kills an estimated 400,000 people annually [3] while *Toxoplasma gondii* causes toxoplasmosis, a disease that can be fatal in immunocompromised people and foetuses [4]. Apicomplexans have a complicated life cycle in which they undergo both asexual and sexual replication as they progress between intermediate and definitive hosts and within the tissues of each host. In both *Plasmodium* and *Toxoplasma*, the mETC is essential in all the life cycle stages studied [2, 5–7] and it is a major drug target, e.g. for the clinically used atovaquone [8–12]. *T. gondii* is an important pathogen, and as it can be cultured in quantities suitable for organelle enrichment and proteomics and is genetically tractable [13], it is a versatile model to study the unique and conserved traits of apicomplexan mETC biology.

The apicomplexan complexes involved in oxidative phosphorylation (**Fig 1B**) are highly divergent from those of mammals. Notably, the genomes of apicomplexan and related organisms [14] lack genes encoding the subunits of complex I (NADH:ubiquinone oxidoreductase), rendering them insensitive to the complex I inhibitor rotenone [15, 16]. Thus, while in mammals electrons are fed from the TCA cycle on NADH to complex I and on succinate to complex II (succinate dehydrogenase), electrons from NADH enter the apicomplexan mETC through an alternative NADH dehydrogenase (NDH2) (**Fig 1B**) [17]. In total, five quinone reductases are suggested to operate in Apicomplexa. In the absence of complex I, complex III (the ubiquinol:cytochrome *c* oxidoreductase, also known as the cytochrome $bc_1$ complex), is the first proton pumping complex in the apicomplexan mETC. Information is already available about the composition of *Toxoplasma* complex IV (the cytochrome *c*:$O_2$ oxidoreductase) and ATP synthase, including the presence of parasite-specific subunits [6, 7, 18]. However, despite the importance of complexes II and III to the function of the apicomplexan mETC, their composition remains uncharacterised.

The structure and function of complex III in the mETC of opisthokonts, such as yeast and mammals, is well understood [19–23]. Three core subunits are directly involved in electron transfer from ubiquinol to the mobile carrier protein, cytochrome *c*, and in the concomitant proton pumping reaction: the cytochrome *b*, Rieske and cytochrome $c_1$ subunits [24]. One electron is passed from ubiquinol bound at the $Q_o$ site, via two haem *b* groups on cytochrome *b*, to ubiquinone at the $Q_i$ site; the other is transferred via the Rieske iron-sulphur (FeS) cluster

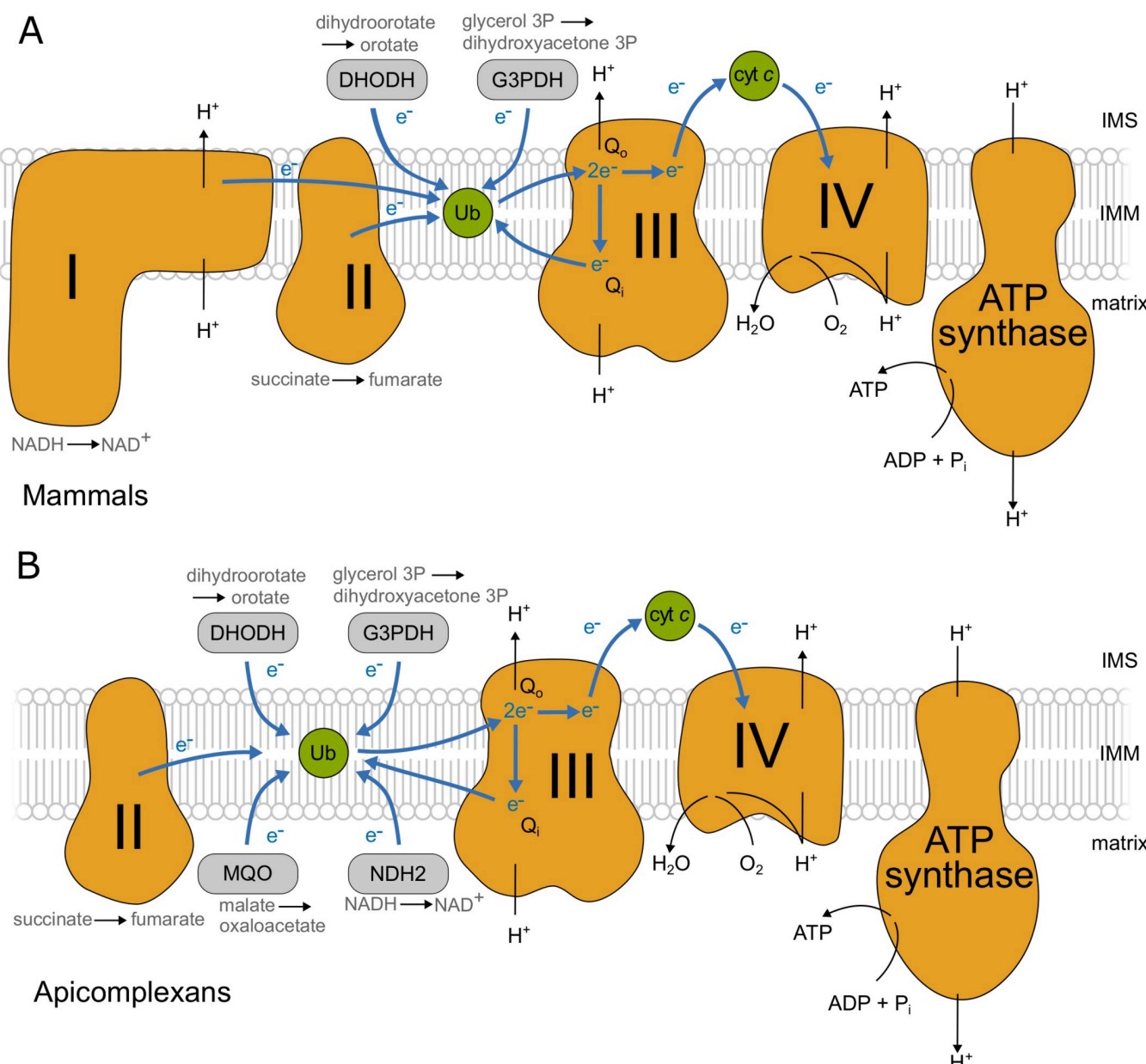

**Fig 1. mETC and $F_1F_o$-ATP synthase in mammals and apicomplexans.** A schematic showing the main features of the mitochondrial electron transport chain (mETC) and ATP synthase in mammals (A) and in apicomplexans (B). The four complexes are depicted in orange (Complex I: NADH:ubiquinone oxidoreductase; Complex II: succinate dehydrogenase; Complex III: cytochrome $bc_1$ complex; Complex IV: cytochrome $c$ oxidase; and $F_1F_o$-ATP synthase). Mobile electron carriers (ubiquinone/ubiquinol and the intermembrane space protein cytochrome $c$) are depicted in green. Main alternative entry points (the dehydrogenases DHODH: dihydroorotate dehydrogenase; G3PDH:glycerol 3-phosphate dehydrogenase; MQO:malate:quinone oxidoreductase and NDH2: type 2 NADH dehydrogenase (*Toxoplasma* possesses two copies)) are depicted in grey. The path of electron flow across the mETC complexes is depicted by blue arrows and the path of protons across the inner mitochondrial membrane (IMM) from the matrix to intermembrane space (IMS) and back is depicted with black arrows.

and cytochrome $c_1$ to the mobile cytochrome $c$ protein. All complexes III are structural dimers, with the Rieske subunit crossing the dimer interface. Each monomer of the mammalian complex contains eight further 'supernumerary' subunits [25], and the yeast complex contains seven [26]. The subunit composition of complex III in apicomplexans is currently unknown. The three key conserved catalytic subunits, the cytochrome $b$, Rieske and cytochrome $c_1$

proteins, along with four well-conserved supernumerary subunits, can be readily identified by homology searches [14, 17, 27]. Further apicomplexan or myzozoan-specific subunits, as have been identified in apicomplexan complex IV and ATP synthase [6, 7, 18], likely remain to be identified.

Despite its important role in parasite survival, and as a drug target, there are still large gaps in our knowledge of the apicomplexan mETC [17, 28]. In this study, we use a complexome profiling proteomic approach to explore the highly divergent mETC and ATP synthase of *T. gondii*. We confirm previously identified subunits of complex IV and ATP synthase, and propose new apicomplexan and myzozoan-specific subunits for those complexes. We investigate in detail the composition of complexes II and III, and describe and validate four previously undescribed structural subunits of complex III. We show that these complex III subunits are essential for parasite growth, for mETC function, and for enzyme integrity. Along with a contemporaneous study [29] this is the first description of the composition of apicomplexan complex III; it increases our understanding of this critical complex and sets the stage for future structural and drug discovery studies.

## Results

### Complexome profiling of *Toxoplasma* mitochondria identifies the respiratory complexes and reveals putative new subunits of complex IV and ATP synthase

Over the last decade complexome profiling has proved a powerful technique for studying mitochondrial macromolecular complexes, providing large scale proteomic data of complexes separated under native conditions [30–33]. Here, we applied complexome profiling to the *Toxoplasma* mitochondrial complexes to gain a fuller understanding of their composition. *Toxoplasma* mitochondria were enriched as described previously [34], and then further isolated via a Percoll gradient purification step (**Figs 2A and S1**). Immunoblot analyses showed the mitochondrial fraction had only a low abundance of other prominent *Toxoplasma* cellular components, such as pellicles and the plasma membrane (**Fig 2B**), although some host (*Chlorocebus sabaeus*) cell mitochondria were also recovered (**S2A Fig**). As observed previously [35] apicoplast contamination was observed in the mitochondrial fraction. Mitochondrial samples were solubilised in 1% *n*-dodecyl β-D-maltoside (DDM) detergent and protein complexes separated by size under native conditions by blue-native PAGE (BN-PAGE) (**S1 Fig**). The gel shown in **S2A Fig** was cut into 63 gel slices and each slice analysed by mass spectrometry. The top two slices were excluded from analysis as they typically contain insoluble aggregates that did not fully enter the gel, thus distorting abundance profiles. Then, the relative abundance of each protein in each of the remaining 61 slices shown in **S2A Fig** was calculated to create the 'complexome profiles' that can be clustered to identify co-migrating proteins that are bound together in complexes of different total molecular mass. In total, 842 *Toxoplasma* proteins were identified (**S1, S2, and S3 Tables**), including 75% (**Fig 2C and S3 Table**) of the 170 proteins assigned to the mitochondrial membrane fraction in a recent organelle proteomics atlas, generated for *Toxoplasma* via Localisation of Organelle Proteins by the Isotope Tagging (LOPIT) method [36]. This comparison suggests that our isolation process captured the majority of mitochondrial membrane proteins. The total also includes 41% of the 229 proteins assigned to the soluble mitochondrial fraction and 46% of the 405 proteins from a *Toxoplasma* mitochondrial matrix proteome [7] (**Fig 2C and S3 Table**). In total, 261 proteins with previous proteomic support for mitochondrial localisation were identified (**S3 Table**). Interestingly, most known subunits of the *Toxoplasma* mitoribosome were not detected and the three

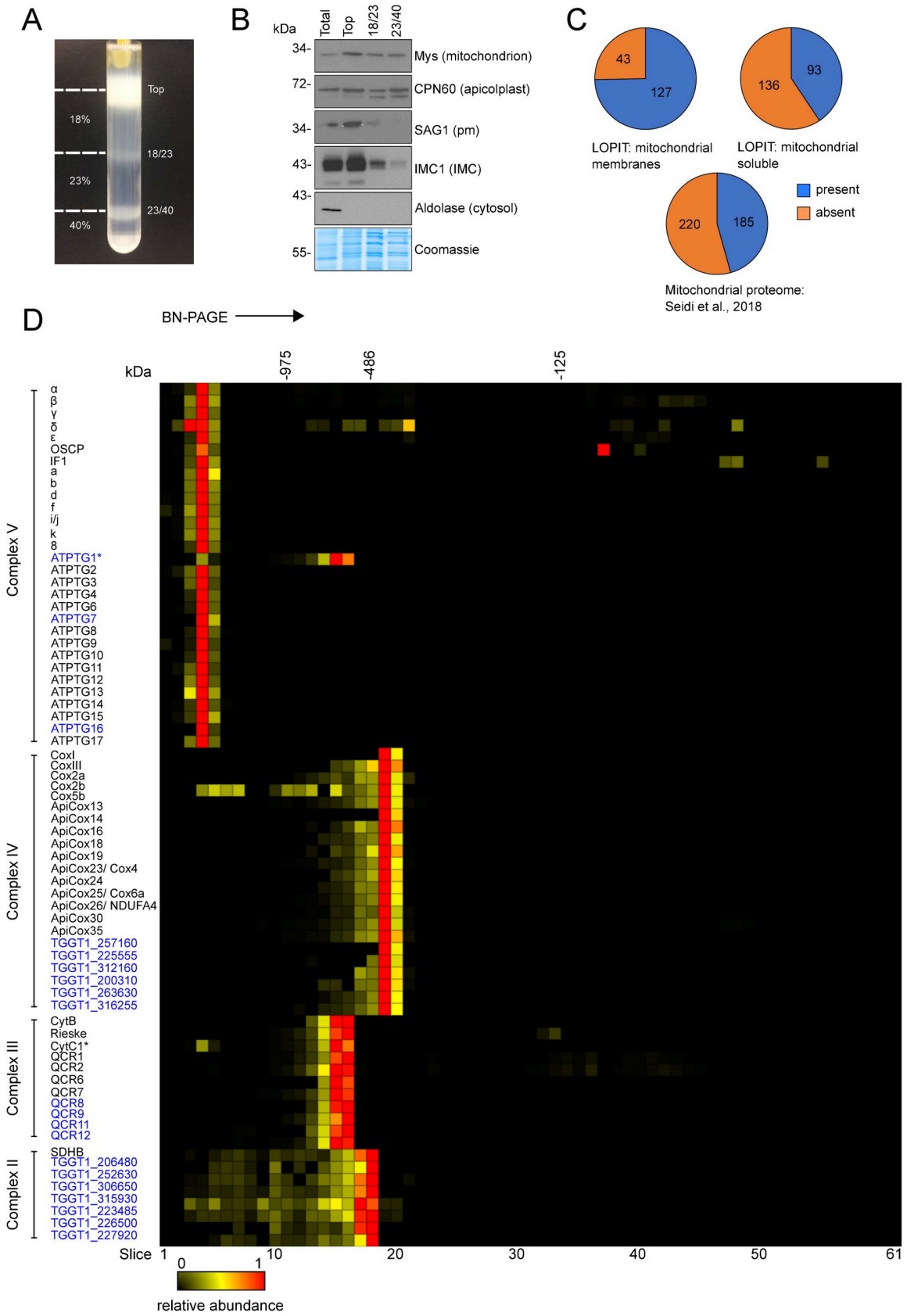

**Fig 2. Complexome profiling of *Toxoplasma* respiratory complexes.** (**A**) Percoll gradient used for density-gradient fractionation. Samples were taken from the top, 18%/23% and 23%/40% Percoll interfaces (marked with dashes lines) for analysis via immunoblot (shown in B). The fraction recovered from the 23%/40% Percoll interface was used for complexome profiling. (**B**) Immunoblot analysis of fractions from total cell lysate (total) and the density-gradient fractionation (top, 18/23 and 23/40) with antibodies against marker proteins for several cell components. Protein concentration from each fraction was quantified with a Bradford assay and 5 µg loaded per lane. Equal loading was confirmed by comassie staining (bottom panel). The following antibodies were used: Mys (mitochondrion), CPN60 (apicoplast), SAG1 (plasma membrane), IMC1 (inner membrane complex) and aldolase (cytosol). (**C**) Proportion of three mitochondrial proteome datasets: LOPIT mitochondrial membrane, LOPIT mitochondrial soluble [36] and mitochondrial matrix proteome [7] found in the complexome profile dataset. (**D**) Complexome profile heatmap of the *Toxoplasma* respiratory complexes subunits. Protein IDs or names are shown on the left of their respective profiles. Previously identified subunits are labelled in black, and putative novel subunits are labelled in blue. Heatmaps x-axis depict the 61 gel slices of the BN-PAGE gel from the top (left) to the bottom (right) of the gel. Molecular weight markers are based on the migrations of mammalian mitochondrial complexes of known size. Red indicates the highest relative abundance (1) and black the lowest (0). Asterisks mark subunits CytC1 of complex III and ATPTG1 of ATP synthase, which are both encoded by the same gene, TGGT1_246540, and therefore have the same profile (see **S4 Fig**). The full complexome profiling dataset is provided in **S1 Table**.

detected did not share abundance profile (**S4 Table**). It is possible that the mitoribosome did not enter the gel under the conditions used.

Proteins in the same intact mETC complex co-migrate in native gels [6, 7, 18, 34] and typically display similar relative abundance profiles. Indeed, the dataset generated here showed that numerous previously assigned mETC and ATP synthase proteins co-migrated in the gel, allowing us to identify the migration positions of the different respiratory complexes (**Figs 2D, S2B and S2C and S1 Table**). Previously uncharacterised proteins with similar abundance profiles as the known subunit proteins are therefore potential new components of the *T. gondii* respiratory complexes. The contamination with host mitochondria served as a useful validation for the PAGE and clustering process–we identified the complexes from *C. sabaeus* (the primate origin of the host cells used to propagate *T. gondii* for this experiment) formed by the expected clusters of subunits, and we were able to use the known [37–39] or Mitoprot-predicted [40] mature masses of several *C. sabaeus* proteins as a calibration to estimate the unknown masses of observed *T. gondii* protein complexes (**S2D Fig and S5 Table**).

The composition of ATP synthase in *Toxoplasma* has recently been studied through immunoprecipitation and *in silico* analyses [6, 18]. 27 of the 29 previously identified *Toxoplasma* ATP synthase subunits [6, 18] were detected in our data. We observed that profiles for the conserved α and β subunits showed two peaks, one of which coincided with other subunits from the *Toxoplasma* complex, and the other with ATP synthase subunits from *C. sabaeus* (**S3 Fig**). Closer inspection found that highly abundant peptides common to both species were skewing the profiles. To overcome this, the α and β protein sequences from both species in the database were truncated to before the first common peptide (**S3 Fig and S6 Table**). After truncation, the profiles for *Toxoplasma* ATP synthase subunits α and β coincided with the profiles of the other ATP synthase subunits (**S3 Fig**) with their relative abundance profiles peaking in gel slice 4 (**Figs 2D and S2D**). Our mass calibration (**S2B Fig and S5 Table**) suggests *Toxoplasma* ATP synthase has a total mass of ~1860 kDa, substantially larger than the 900 kDa complex suggested by Huet et al, 2018 [6] and the 1–1.2 MDa complex suggested by Salunke et al, 2018 [18]. Differences from previous studies may be a result from the use of different detergents, which affect the Native migration of complexes. Native PAGE carried out under the same conditions as for the complexome analysis, followed by immunoblotting with an antibody against the ATP synthase β subunit, confirmed the migration of the *Toxoplasma* ATP synthase and its high molecular mass, >1 MDa (**S4A Fig**). Of the known ATP synthase subunits, only OSCP (TGGT1_284540) exhibits an apparent additional peak elsewhere in the gel (slice 37, ~86 kDa), possibly representing an assembly intermediate or electrophoretic degradation. Three additional proteins, TGGT1_246540, TGGT1_290710 and TGGT1_211060 exhibit similar relative abundance profiles to known ATP synthase subunits, suggesting they could be novel

members of this complex (**Figs 2D and S2D**). Indeed, these three proteins were identified in a recent structure of *Toxoplasma* ATP synthase [41] and were named ATPTG1, ATPTG7 and ATPTG16, respectively. Homology searches with the HMMER tool suggests they have a similar phylogenetic distribution to recently discovered myzozoan-restricted ATP synthase subunits [6, 18] (**S4B Fig and S7 Table**), highlighting the divergence of the ATP synthase complex in this lineage. Of note, the TGGT1_246540 gene that encodes ATPTG1 also encodes the cytochrome $c_1$ subunit of complex III (see below). Supporting this, we observe two peaks for the abundance of TGGT1_246540 in the gel: its highest relative abundance is in slice 15, with other complex III subunits, however it also exhibits a peak in slice 4, with ATP synthase subunits. Peptides with a peak in slice 4 (ATP synthase) map to the N-terminal portion of the protein, while peptides with their peak in slice 15 (complex III) map to the C-terminal portion, after residue 153 (**S4C Fig**). Apicomplexan homologs of ATPTG1 in *Hammondia hammondi*, *Babesia bovis*, *Plasmodium falciparum* and *Plasmodium berghei* also contain cytochrome *c1* domains (**S4B Fig and S7 Table**), but homologs from *Cryptosporidium muris*, *Vitrella brassicaformis* and *Perkinus marinus* do not contain this domain.

All of the 14 recently identified nuclear encoded complex IV subunits [7, 34] were identified and co-migrated on the gel, with peak abundance observed in gel slice 19 (**Figs 2D, S2C and S2D**). Using the sequence of the *T. gondii* mitochondrial genome from a recent study [42], we also detected the two mitochondrially encoded complex IV subunits, CoxI and CoxIII (**Fig 2D**), which have not been previously detected in large-scale proteomic studies. The *Toxoplasma* complex IV is estimated to have a total mass of ~460 kDa (**S5 Table**), smaller than the 600 kDa estimated by Seidi et al., 2018 [7], but substantially larger than the 200 kDa of mammals and yeast [43, 44]. The sum of the protein masses of the 16 previously identified subunits, after removal of any mitochondrial targeting sequences estimated by Mitoprot [40], is 364 kDa (**S8 Table**), lower than both experimental estimates, suggesting the presence of additional, unidentified *Toxoplasma* complex IV subunits. In support of this prediction six other proteins (TGGT1_257160; TGGT1_225555; TGGT1_312160; TGGT1_200310; TGGT1_263630; TGGT1_316255) co-migrated with complex IV, suggesting that they may be novel complex IV subunits (**Fig 2D**). The identification of TGGT1_257160 was based on a single peptide with a moderate score (above the $p < 0.05$ threshold but not above the $p < 0.01$ threshold). The sum of their masses is 74 kDa, which, when added to 364 kDa for the known subunits, gives a total of 438 kDa, close to the experimental estimate of 460 kDa. Further evidence for their assignment as complex IV subunits is apparent from close examination of proteomic data from two recent studies [7, 34]; both studies observed these proteins, but did not assign them to complex IV. In addition, the LOPIT dataset [36] suggests all six proteins are present in mitochondrial membranes (**S8 Table**) and homology searches with the HMMER tool suggests their phylogenetic distribution is limited to apicomplexans and coccidians (**S5 Fig and S6 Table**).

The identification of all previously known complex IV subunits and most known ATP synthase subunits both validates our approach and confirms the compositions of these two complexes. Our approach has further highlighted three additional ATP synthase subunits and six candidate new components of complex IV.

## Identification of an unusually large complex II and assignment of new apicomplexan complex II subunits

The composition of the apicomplexan complex II has not previously been investigated. The mammalian complex is made up of four subunits [45], and genes for the soluble domain (FAD and FeS-containing subunits SDHA and SDHB) are readily identified in *T. gondii* [14, 17, 27]. However, the sequences for the membrane-bound subunits responsible for quinone binding

and reduction remain elusive (**Tables 1 and S9**). In our complexome analysis, we found the FeS-containing subunit B (SDHB) to migrate at ~500 kDa, with its highest abundance in slice 18 (**Figs 2D, S2C and S2D and S5 Table**). This size is much larger than for the mammalian and yeast complexes (~130 kDa) [37, 43], but in line with a the similarly large complex of 745 kDa recently proposed for the apicomplexan *Eimeria tenella* [46]. The SDHA subunit profile showed two peaks, one of which coincided with SDHB from *Toxoplasma*, and the other with the known *C. sabaeus* complex II subunits. As described above for α and β subunits of ATP synthase, where we encountered a similar issue, we attempted to overcome this problem by truncating the protein sequences in the database before the common peptides. However, this time the peak relative abundance for the *Toxoplasma* SDHA subunit remained in the same slice as that for the *C. sabaeus* complex. To confirm the migration of SDHB and the apparent large molecular weight of the *T. gondii* complex II, we attached a C-terminal triple hemagglutinin (HA) epitope tag to the SDHB subunit, by endogenous tagging using CRISPR-mediated homologous recombination in the TATiΔ*ku80* parental strain [47] (**S6A Fig**). Clonal lines were isolated by serial dilution and confirmed by PCR (**S6B Fig**). Migration of *Toxoplasma* SDHB-HA on SDS-PAGE confirmed its predicted size from the gene model (TGGT1_215280, ~42 KDa) (**Fig 3A**). BN-PAGE and immunoblot analysis indicated the SDHB subunit migrates between the 480 and 720 kDa marker (**Fig 3B**), consistent with the ~500 kDa observed in our complexome profile. Further, *Toxoplasma* SDHB-HA co-localised with the mitochondrial marker TOM40 [48] in an immunofluorescence assay (IFA), confirming its predicted mitochondrial location (**Fig 3C**). These data suggest that *Toxoplasma* possesses a complex II that is greater in size than in other well studied systems. While the large size could partly be explained by a higher order oligomer of the complex, it may also point to additional parasite-specific subunits. Likewise, functionally, it is expected that there must be at least one other subunit associated in addition to the identifiable SDHA and B subunits, to allow electron transfer from the FeS clusters to ubiquinone. In support of these predictions, seven other proteins were found to co-migrate with SDHB, with peak abundances in gel slice 18 (**Fig 2D**) suggesting the existence of novel complex II subunits. Five of these were identified by the *Toxoplasma* LOPIT study as proteins of the mitochondrial membrane fraction [36], consistent with the expected mitochondrial membrane localisation for complex II (**Tables 1 and S9**); all seven proteins are annotated as hypothetical proteins and do not contain any protein features indicative of function.

To investigate these complex II subunit candidates further we considered their phylogenetic distribution using the HMMER similarity search tool (**Fig 3D and S7 Table**). All of them, except for TGGT1_227920, are well conserved in the apicomplexans *Hammondia hammondi*, *Babesia bovis*, *Plasmodium falciparum* and *Plasmodium berghei* (**Fig 3D**), although absent from ciliates, suggesting myzozoa restricted distributions. To discriminate between putative complex II subunits and other mitochondrial proteins we used the fact that complex II is present in *Cryptosporidium muris*, but absent from *Cryptosporidium parvum* [17]. Therefore, complex II subunits should follow this phylogenetic pattern, as observed for SDHA and SDHB (**Fig 3D**). Four of the seven co-migrating proteins (TGGT1_306650, TGGT1_252630, TGGT1_206480 and TGGT1_315930) follow this same pattern, supporting their allocation to complex II. These four proteins were also identified in the recent *Toxoplasma* mitochondrial matrix proteome [7]. In summary, we identify seven candidate novel *Toxoplasma* complex II subunits by co-migration and find phylogenetic support for four of them (**Tables 1 and S7**).

In addition to complex II, we observed evidence for the following quinone reductases: two single-subunit type II NADH dehydrogenases (NDH2) (in *Toxoplasma* NDH2-I and II [49]), malate:quinone oxidoreductase (MQO), dihydroorotate dehydrogenase (DHODH) and FAD-dependent glycerol 3-phosphate dehydrogenase (G3PDH) [17] (**S7 Fig**). These proteins all

**Table 1.**

**Details of putative complex II subunits.** Phenotype scores were taken from Sidik et al., 2016 [56]; Matrix proteome from Seidi et al., 2018 [7] (Y–present in proteome, N–absent); LOPIT from Barylyuk et al., 2020 [36] (mm–mitochondrial membrane, ms- mitochondrial soluble N/A–not identified in dataset). TMD: Transmembrane domain, predicted by TMHMM.

| Gene ID | Name | Phenotype score | Matrix proteome | LOPIT | Length (aa) | Size (kDa) | TMD |
|---|---|---|---|---|---|---|---|
| TGGT1_215590 | SDHA | -3.96 | Y | ms | 669 | 72.7 | No |
| TGGT1_215280 | SDHB | -2 | Y | mm | 342 | 38.6 | No |
| TGGT1_206480 | hypothetical | -1.8 | Y | mm | 207 | 22.5 | No |
| TGGT1_252630 | hypothetical | -1.45 | Y | mm | 136 | 15.1 | No |
| TGGT1_306650 | hypothetical | -1.8 | Y | mm | 286 | 31.4 | No |
| TGGT1_315930 | hypothetical | -3.84 | Y | mm | 157 | 18.1 | No |
| TGGT1_223485 | hypothetical | -1.49 | N | N/A | 93 | 10.3 | No |
| TGGT1_226500 | hypothetical | -0.89 | N | mm | 92 | 11.1 | Yes (1) |
| TGGT1_227920 | hypothetical | -2.42 | N | N/A | 85 | 9.5 | No |

migrated to positions under 140 kDa, consistent with their presence as monomers or dimers, and their predicted sizes from current gene models.

## Identification of putative new apicomplexan complex III subunits

The composition of the apicomplexan complex III has not previously been studied. Six canonical subunits of eukaryotic complex III can be readily identified in apicomplexan genomes, including *Toxoplasma*, by homology searches [7, 14, 17, 27] (**Tables 2 and S9**). They include the catalytic Rieske and cytochrome $c_1$ subunits (TGGT1_320220 and TGGT1_246540), herein named Rieske and CytC1, and four supernumerary subunits (TGGT1_236210; TGGT1_202680; TGGT1_320140 and TGGT1_288750) herein named QCR1, QCR2, QCR6 and QCR7. These six conserved subunits co-migrated with peak abundance in gel slice 15, corresponding to ~670 kDa, slightly larger than the 500 kDa complex found in mammals [39, 43] (**Figs 2D, S2B, S2C, and S2D and S5 Table**). We also detected the mitochondrially encoded subunit, cytochrome *b* (CytB) identified with a single peptide with a moderate score (above the $p<0.05$ threshold but not above the $p<0.01$ threshold) (**Fig 2D**). Three other subunits found in the mammalian complex (UQCRQ, UQCR10 and UQCR11) are not identified in the *Toxoplasma* genome by homology searches. The larger size of *T. gondii* complex III, and the inability to identify homologs for these three subunits, suggest further subunits of complex III are present. In line with this prediction, six additional proteins co-migrate with the known complex III subunits in our complexome analysis (**Fig 2D and S1 Table**). Given the importance of complex III as a drug target, and the fact that little is known about the apicomplexan complex, we proceeded to provide validation to the assignment of these putative new subunits as integral parts of *Toxoplasma* complex III.

All 13 proteins identified here as potential complex III subunits are detected in the LOPIT dataset [36], 12 of them in the mitochondrial membranes fraction (**Tables 2 and S9**). The only protein not predicted to be mitochondrial is encoded by TGGT1_233220; it was predicted to be Golgi and/or plasma membrane localised, and so was not studied further. Of the remaining 12 proteins, 11 were also identified in a *Toxoplasma* mitochondrial matrix proteome generated through proximity tagging (**Tables 2 and S9**) [7]. The exception is the highly conserved CytB subunit, either due to it being a highly hydrophobic, membrane-embedded protein or to previously uncertain gene-model annotation.

The mETC chain is essential for *Toxoplasma* survival in culture [6, 7, 17], and inhibitors of complex III result in parasite death [50–55]. Consequently, subunits of complex III are predicted to be essential for growth in culture. A genome wide CRISPR screen has provided

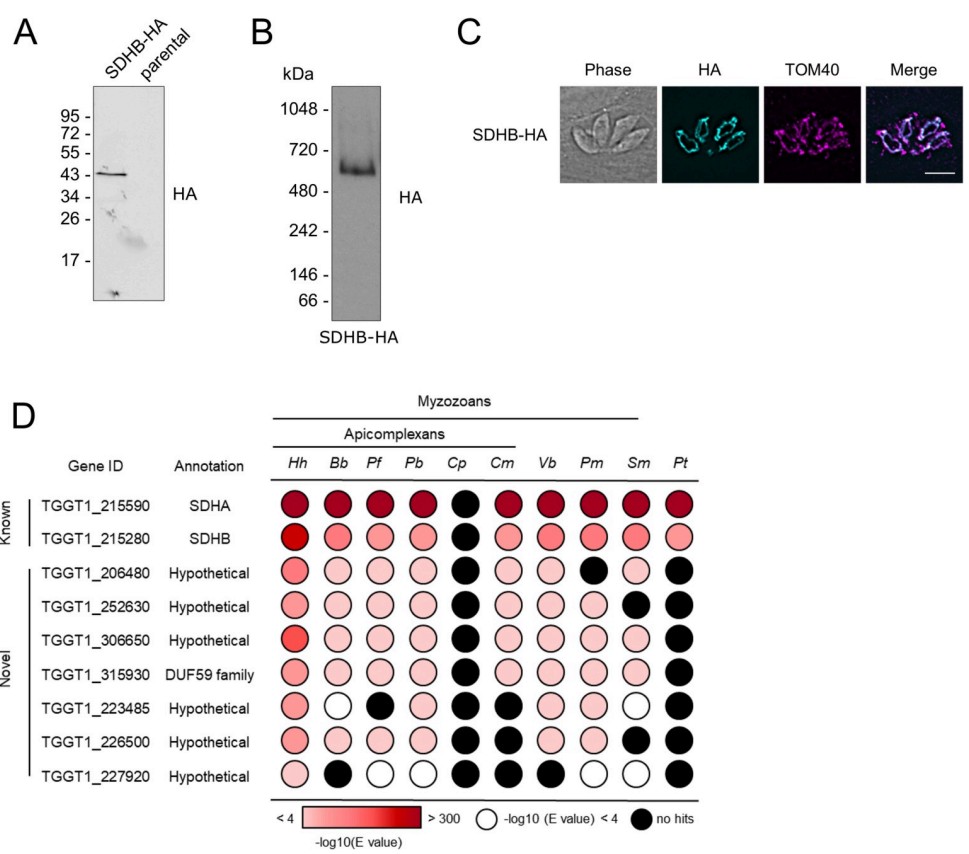

**Fig 3. Analysis of putative *Toxoplasma* complex II subunits. (A)** Immunoblot analysis of complex II subunit SDHB endogenously tagged with an HA epitope. Proteins from total lysate were separated by SDS-PAGE and detected using anti-HA antibodies. Parasites from the parental line (TATIΔ*ku80*) were analysed as negative control. **(B)** Total lysate from SDHB-HA separated by BN-PAGE and immunolabelled with anti-HA antibodies. **(C)** Immunofluorescence assay with parasites expressing the endogenously HA-tagged SDHB (cyan), showing co-localisation with the mitochondrial marker protein TOM40 [48] (magenta), along with merge and phase. Scale bar is 5 μm. **(D)** Table showing the previously predicted (known) and complexome identified putative novel (novel) complex II subunits and their homology distribution across key groups. Homology searches were performed using the HMMER tool [57]. Coloured circles refer to the e-value from the HMMER search: white indicates a hit with an e-value above 0.0001, black indicates no hits, and red indicates hits with an e-value below 0.0001, as indicated in the coloured scale. Full data are given in **S7 Table**. *Hh*: *Hammondia hammondi*; *Bb*: *Babesia bovis*; *Pf*: *Plasmodium falciparum*; *Pb*: *Plasmodium berghei*; *Cp*: *Cryptosporidium parvun*; *Cyryptosporidium muris*; *Vb*: *Vitrella brassicaformis*; *Pm*: *Perkinsus marinus*; *Sm*: *Symbiodinium microadriaticum*; *Pt*: *Paramecium tetraurelia*.

phenotype scores for thousands of genes [56]. 10 of the 11 nuclear-encoded genes encoding putative complex III subunits have phenotype scores below -3 (**Table 2**), indicating a high degree of contribution to fitness. In contrast, TGGT1_297160, encoding one of the hypothetical proteins, has a high phenotypic score (0.1), suggesting a dispensable role for growth in culture. In addition, we considered the predicted protein size. By considering the well-studied complex III of mammals, in which the remaining subunits, UQCRQ, UQCR10 and UQCR11, are all smaller than 150 amino acids, we hypothesised that any new *Toxoplasma* complex III subunits would be of similar size. While four of the five hypothetical complex III subunit candidates are under 150 amino acids in length, TGGT1_297160 is predicted to encode a 564 amino acid protein (**Table 2**). The high fitness score and large size render TGGT1_297160 an unlikely candidate for a novel complex III subunit. These bioinformatic data allowed us to focus on four putative novel subunits: TGGT1_201880, TGGT1_214250, TGGT1_207170 and TGGT1_227910.

**Table 2. Details of nuclear encoded complex III subunits.** Phenotype scores were taken from Sidik et al., 2016 [56]; Matrix proteome from Seidi et al., 2018 [7]; (Y–present in proteome, N–absent); LOPIT from Barylyuk et al., 2020 [36] (mm–mitochondrial membrane, N/A–not identified in dataset). TMD: Transmembrane domain, predicted by TMHMM server. TGGT1_297160 was initially identified as a potential complex III subunit but was later excluded.

| Gene ID | Name | Human/yeast homolog | Phenotype score | Matrix proteome | LOPIT | Length (aa) | TMD (TMHMM) |
|---|---|---|---|---|---|---|---|
| TGGT1_246540 | CytC1 | CYC1/CYT1 | -4.36 | Y | mm | 398 | Yes (1) |
| TGGT1_320220 | Rieske | UQCRFS1/RIP1 | -5.76 | Y | mm | 487 | No |
| TGGT1_236210 | QCR1 | UQCRC1/QCR1 | -4.74 | Y | mm | 509 | No |
| TGGT1_202680 | QCR2 | UQCRC2/QCR2 | -4.3 | Y | mm | 563 | No |
| TGGT1_320140 | QCR6 | UQCRH/QCR6 | -3.69 | Y | mm | 89 | No |
| TGGT1_288750 | QCR7 | UQCRB/QCR7 | -4.04 | Y | mm | 234 | No |
| TGGT1_227910 | QCR8 | UQCRQ/QCR8 | -3.2 | Y | mm | 122 | No |
| TGGT1_201880 | QCR9 | UQCR10/QCR9 | -3.63 | Y | mm | 128 | No |
| TGGT1_214250 | QCR11 | - | -3.94 | Y | mm + o | 80 | Yes (1) |
| TGGT1_207170 | QCR12 | - | -3.44 | Y | mm | 141 | Yes (1) |
| TGGT1_297160 | - | - | 0.1 | Y | mm | 564 | No |

Using HHPRED [57] to determine structural similarity, two of our candidates had regions predicted to be homologous to existing complex III subunit structures. TGGT1_227910 displayed homology with *Bos taurus* UQCRQ/QCR8 (97.51% probability, e-value 2.2e-6, between amino acids 58–119, PDB annotation: 1PP9_T). TGGT1_201880 had homology with *Saccharomyces cerevisiae* cytochrome $bc_1$ complex subunit 9 (QCR9) (99.82% probability, e-value 1.7e-23, between amino acids 71–128, PDB annotation: 3CX5_T). Therefore, these proteins are renamed QCR8 (TGGT1_227910) and QCR9 (TGGT1_201880). TGGT1_214250, and TGGT1_207170 did not display strong homology with any other known proteins; they are thus predicted to be parasite-specific and were named QCR11 and QCR12 respectively.

We then investigated the conservation of complex III subunits across different species. The CytC1 and Rieske subunit genes were widely distributed and found across eukaryotes in all groups with the exception of *Cryptosporidium parvum*, which lacks complex III [17] (**Fig 4A**). QCR1 and QCR2 were likewise widely conserved but were also conserved in *Cryptosporidium parvum*. This is likely due to their peptidase M16 domains, which are also found in mitochondrial processing peptidase (MPP) proteins in *Cryptosporidium parvum* and which may play a role in mitochondrial protein cleavage. QCR6 and QCR7 are widely conserved, although homologs were not detected in all groups, possibly due to the small protein size and sequence divergence. Homologs for QCR8 and QCR9 could not be detected outside the myzozoa, likely due to high sequence divergence. Interestingly, QCR11 appeared only present in myzozoans and QCR12 was restricted to the coccidia.

Proteomic data, phenotype scores and structural prediction supported the assignment of 11 proteins, including 4 new ones, from the complexome analysis as complex III subunits. A parallel independent study which characterised the *Toxoplasma* complex III via co-immunoprecipitation [29] identified the same 11 subunits, providing support to these assignments. This new subunit nomenclature was agreed with Hayward and colleagues to keep uniformity in the field.

### Putative complex III subunits localise to the mitochondrion and are part of a ~670 kDa complex

Our complexome profiling and bioinformatic analysis identified 10 nuclear encoded proteins, and one mitochondrially encoded protein of complex III. To confirm the predicted mitochondrial localisation of the nuclear encoded subunits, proteins were C-terminally tagged with a

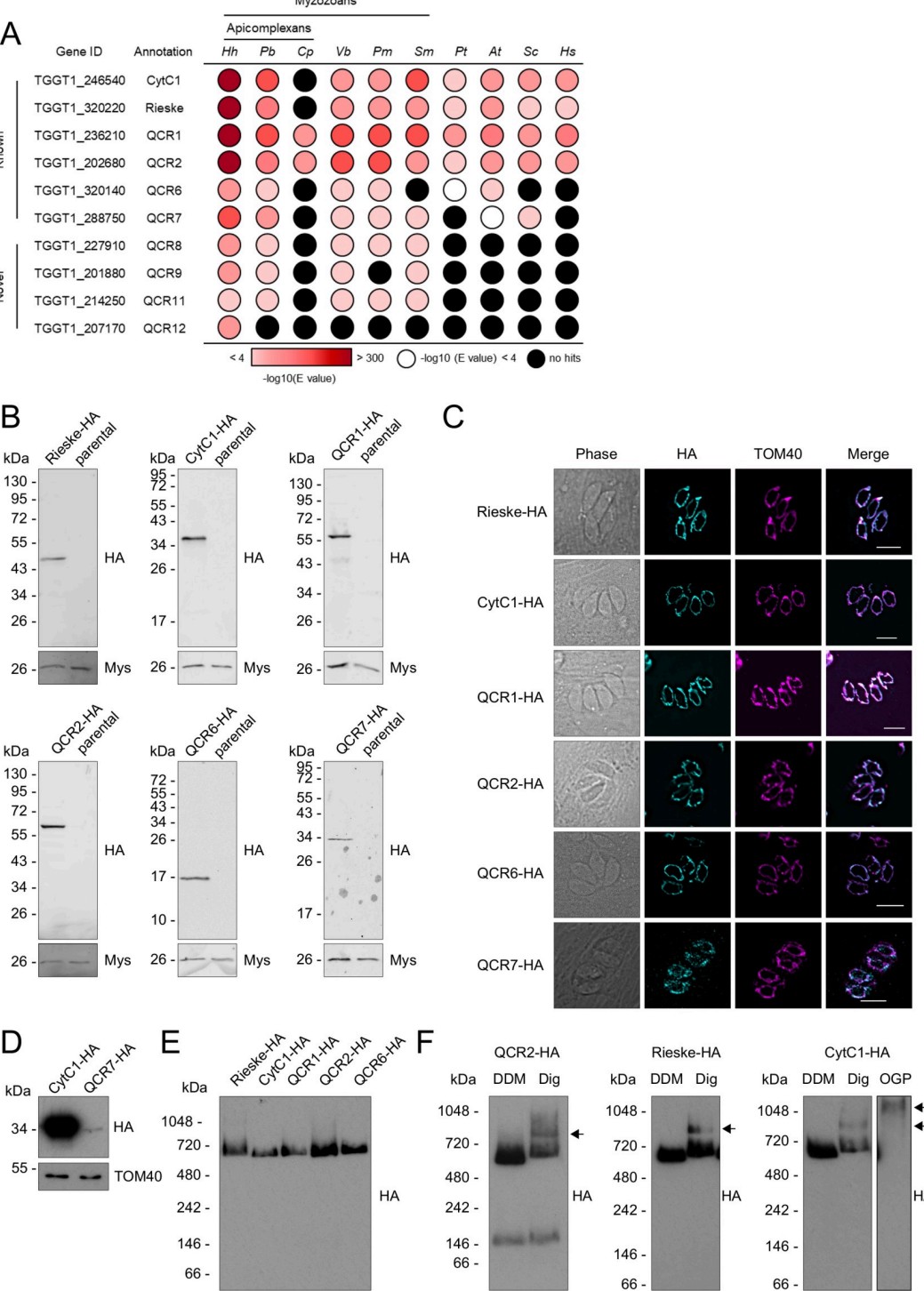

**Fig 4. Analysis of putative *Toxoplasma* complex III subunits. (A)** Table showing the previously predicted (known) and complexome identified putative novel (novel) complex III subunits and their homology distribution across key groups. Homology searches were performed using the HMMER tool [57]. Coloured circles refer to the e-value from the HMMER search: white indicates a hit with an e-value above 0.0001, black indicates no hits, and red indicates hits with an e-value below 0.0001, as indicated in the coloured scale. Full data are given in **S7 Table**. *Hh*: *Hammondia hammondi*; *Pb*: *Plasmodium berghei*; *Cp*: *Cryptosporidium parvun*; *Cyryptosporidium muris*; *Vb*: *Vitrella brassicaformis*; *Pm*: *Perkinsus marinus*; *Sm*: *Symbiodinium microadriaticum*; *Pt*: *Paramecium tetraurelia*; *At*: *Arabidopsis thaliana*; *Sc*: *Saccharomyces cerevisiae*; *Hs*: *Homo sapiens*. **(B)** Immunoblot analysis of putative complex III subunits endogenously tagged with an triple-HA epitope tag.

Proteins from total lysate were separated by SDS-PAGE, blotted and detected using anti-HA antibodies. Parasites from the parental line (Δ*ku80*) were analysed as negative control. Antibodies against the mitochondrial protein Mys (TGME49_215430) [84] were used as a loading control. **(C)** Immunofluorescence assay of parasites with endogenously HA-tagged putative complex III subunits (cyan), showing co-localisation with the mitochondrial marker protein TOM40 (magenta), along with merge and phase. Scale bars are 5 μm. **(D)** Immunoblot analysis of complex III subunits CytC1-HA and QCR7-HA. Proteins from total lysate were separated by SDS-PAGE, blotted and detected using anti-HA antibodies. Antibodies against the mitochondrial protein TOM40 [48] were used as a loading control. **(E)** Total lysate from tagged complex III subunits separated by BN-PAGE, blotted and immunolabelled with anti-HA antibodies. **(F)** Total lysate from tagged complex III subunits, solubilised with different detergents, DDM or digitonin, separated by BN-PAGE, blotted and immunolabelled with anti-HA antibodies.

triple HA epitope tag, as described above (**S6A Table**). Clonal lines were isolated by serial dilution and confirmed by PCR (**S6B Fig**). We successfully endogenously tagged the CytC1, Rieske, QCR1, QCR2, QCR6 and QCR7 proteins. Expression of HA tagged proteins was confirmed by immunoblot and immunofluorescence (**Fig 4B and 4C**). QCR7 appeared to have a much weaker immunoreactive signal (**Fig 4D**), perhaps suggesting lower expression levels, or the HA epitope tag may interfere with recruitment into the complex, thereby leading to its turnover. All candidates co-localised with the mitochondrial marker protein TOM40 [48], confirming their mitochondrial localisation (**Fig 4C**).

To validate the predicted ~670 kDa size of complex III, lysates from each epitope tagged line except QCR7 were separated by BN-PAGE and immunoblotted with antibodies against HA. All putative subunits co-migrated below the 720 kDa molecular weight marker, consistent with the expected ~670 kDa mass (**Fig 4E**). A lower molecular weight band, of ~150 kDa, was observed in some immunoblots of QCR2 (**Fig 4E**), possibly representing an assembly intermediate or electrophoretic degradation.

In other eukaryotes, mETC complexes can associate together to form respiratory supercomplexes [43, 58, 59]. These can be detected by native PAGE upon solubilisation by mild nonionic detergents, such as digitonin [43]. However respiratory supercomplex formation has not previously been observed in *Toxoplasma*. In order to investigate this, lysates from epitope tagged Rieske, CytC1 and QCR2 parasites were solubilised in DDM, digitonin or octyl β-D-glucopyranoside (OGP) and separated by BN-PAGE. In digitonin solubilised samples, compared to samples solubilised in DDM, a second, higher molecular weight band was observed between the 720 and 1048 kDa molecular weight markers, suggesting complex III's participation in a supercomplex (**Fig 4F**). Proteins from total cells of the CytC1-HA line solubilised with OGP displayed an even higher molecular weight band, above 1048 kDa.

## Novel complex III subunits are essential for parasite growth, mitochondrial membrane potential and intact complex III

To further study the function of QCR8, QCR9, QCR11 and QCR12 we generated conditional knockdown lines of each gene by replacing the native promoter with an anhydrotetracycline (ATc) regulated promoter, as previously described [34] (**S8A Fig**). The knockdowns were generated in a parental line in which the Rieske complex III subunit was endogenously tagged with a triple HA epitope tag. Integration of the regulated promoter was confirmed by PCR (**S8B Fig**) and downregulation of transcripts after addition of ATc confirmed by qRT-PCR (**S8C Fig**). Depletion of each of the four genes resulted in a severe parasite growth defect in plaque assays (**Fig 5A**) demonstrating their importance for parasite fitness.

It is expected that disruption of complex III would result in disruption of the mitochondrial membrane potential, as observed when complex III is inhibited by the drug atovaquone [60, 61]. We were able to observe the membrane potential of all four of the subunit conditional knockdown mutants using the cationic carbocyanine dye, JC-1, for which membrane potential

dependent accumulation in mitochondria result in change from green to red fluorescence [62]. The control treatment of parasites with valinomycin, an ionophore that abolishes the mitochondrial membrane potential, as well as the complex III inhibitors atovaquone and antimycin a, resulted in a decrease of the red signal population, indicating a loss of membrane potential, as expected. Likewise, depletion of each of the four subunits through treatment with ATc, resulted in disruption of the membrane potential, observed as a decrease in the red signal population (**Fig 5B**). Treatment of the parental line, TATi*Δku80*, with ATc did not result in any such decrease.

In *Toxoplasma*, changes in mitochondrial morphology have been observed in response to major mitochondrial dysfunction [34]. We assessed the impact of downregulation of QCR8, QCR9, QCR11 and QCR12 on mitochondrial morphology via immunofluorescence microscopy. No significant defect in mitochondrial morphology was observed after downregulation (**S10 Table**), suggesting the absence of general mitochondrial biogenesis defect and pointing to specific defect in complex III.

The putative structural role of supernumerary subunits suggest that their depletion may result in stalled assembly or impaired stability of the complex, as shown in *S. cerevisiae* [63, 64] and in other *Toxoplasma* mETC complexes [6, 7]. Using immunoblotting against the endogenously tagged Rieske subunit, we assessed the migration of complex III in BN-PAGE following depletion of each of the four components (**Fig 6A, 6B, 6C and 6D**). Depletion of all four subunits resulted in loss of the complex. This was apparent in QCR8, QCR11 and QCR12 after two days of downregulation and in QCR9 after three days. Addition of ATc to the parental Rieske-HA line, where no promoter replacement was performed, had no effect on complex III stability (**Fig 6E**). These data suggest all four proteins are required for complex formation and/or stability. Moreover, at these time points, two unrelated mitochondrial complexes, the mitochondrial protein import complex Translocon of the Outer mitochondrial Membrane (TOM) (**Fig 6F**) and the mETC complex IV (**Fig 6G**) were observed to remain fully formed confirming the specificity of the effect on complex III.

Our data demonstrate the existence of four novel complex III subunits that are crucial for the formation and/or stability of complex III. Disruption of any of them results in loss of the complex, collapse of the membrane potential and lack of parasite viability.

## Discussion

The mETC and $F_1F_o$-ATP synthase of apicomplexan parasites are highly divergent from those of their mammalian hosts [6, 7, 17, 18]. Beyond energy metabolism [5–7], the mETC plays a role in pyrimidine biosynthesis [2], making it indispensable at every stage of the apicomplexan life cycle studied to date. Our complexome analysis has provided a comprehensive overview of the apicomplexan mETC and ATP synthase complexes, identifying 71 protein subunits including 20 previously unknown subunits, and describing the composition of complex II and complex III for the first time in an apicomplexan model.

Isolating mitochondria from *Toxoplasma* has proven challenging. Here, we utilised a Percoll density gradient purification step, but the resulting fraction contained both mitochondria and apicoplasts, and had some contamination from other cell components. Previously, other techniques have also been unable to obtain pure fractions [18, 34, 36], highlighting the difficulty in obtaining pure cellular fractions from *Toxoplasma*. Despite this, our highly enriched fraction was suitable for proteomic analysis by complexome profiling. Host cell mitochondria were also present in the final sample. However, the mammalian mitochondrial complexes are well characterised, and we exploited their presence in the sample for more precise calibration of the masses of *Toxoplasma* mitochondrial complexes.

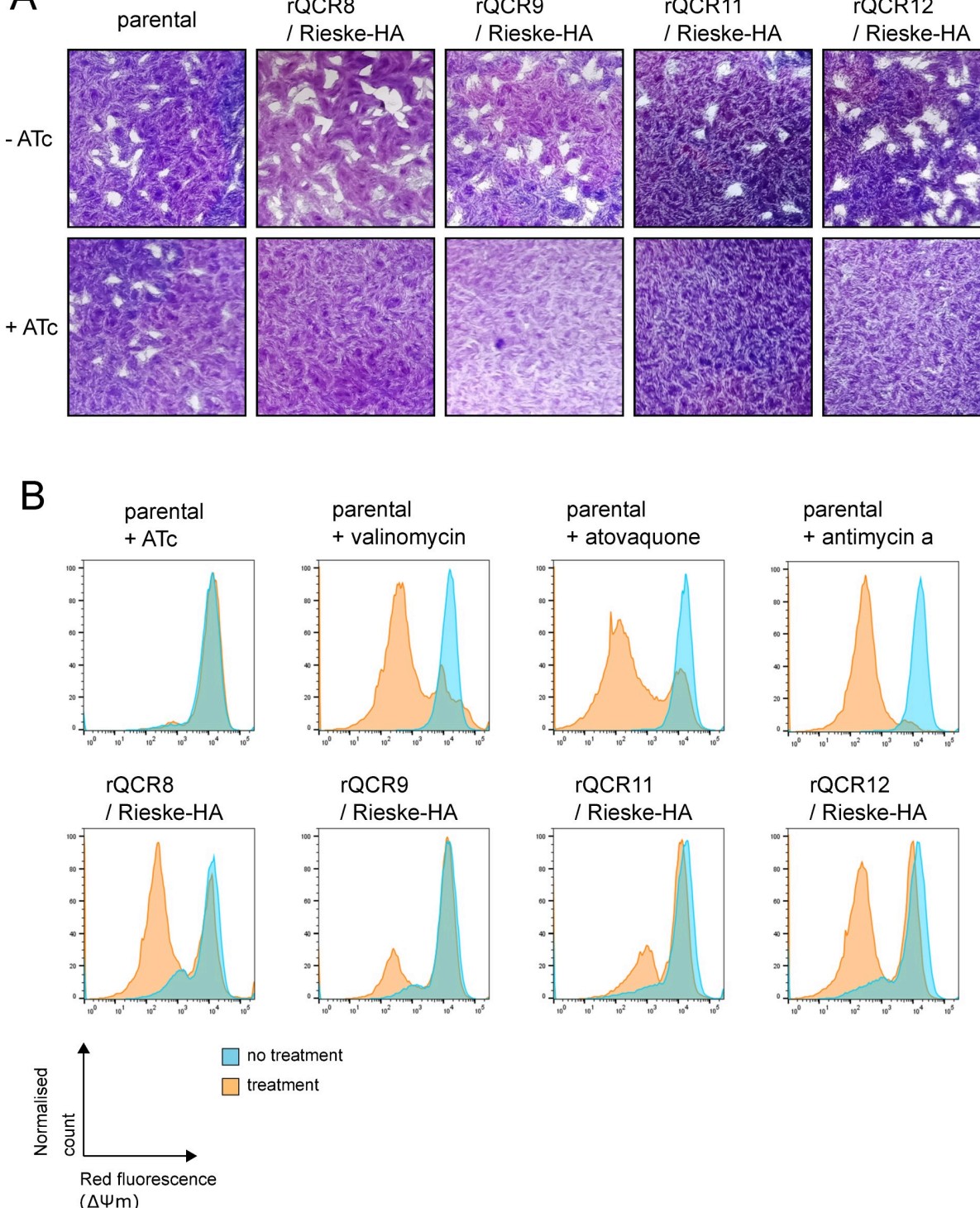

**Fig 5. Loss of the new complex III subunits affects parasite growth and mitochondrial membrane potential. (A)** Plaque assays of the parental line (parental) and four promoter replacement lines (rQCR8/9/11/12 / Rieske-HA). Parasites were grown for 9–10 days in the absence (-ATc) or presence of ATc (+ATc) before being fixed and stained with crystal violet. **(B)** Mitochondrial membrane potential detected using the JC-1 probe of parental (parental) and each of the four promoter replacement lines (rQCR8/9/11/12 / Rieske-HA) in the absence (blue) or presence (orange) of ATc, or valinomycin/atovaquone/antimycin a. Representative histograms of red fluorescence, which is dependent on mitochondrial membrane potential, recorded by flow cytometry are shown.

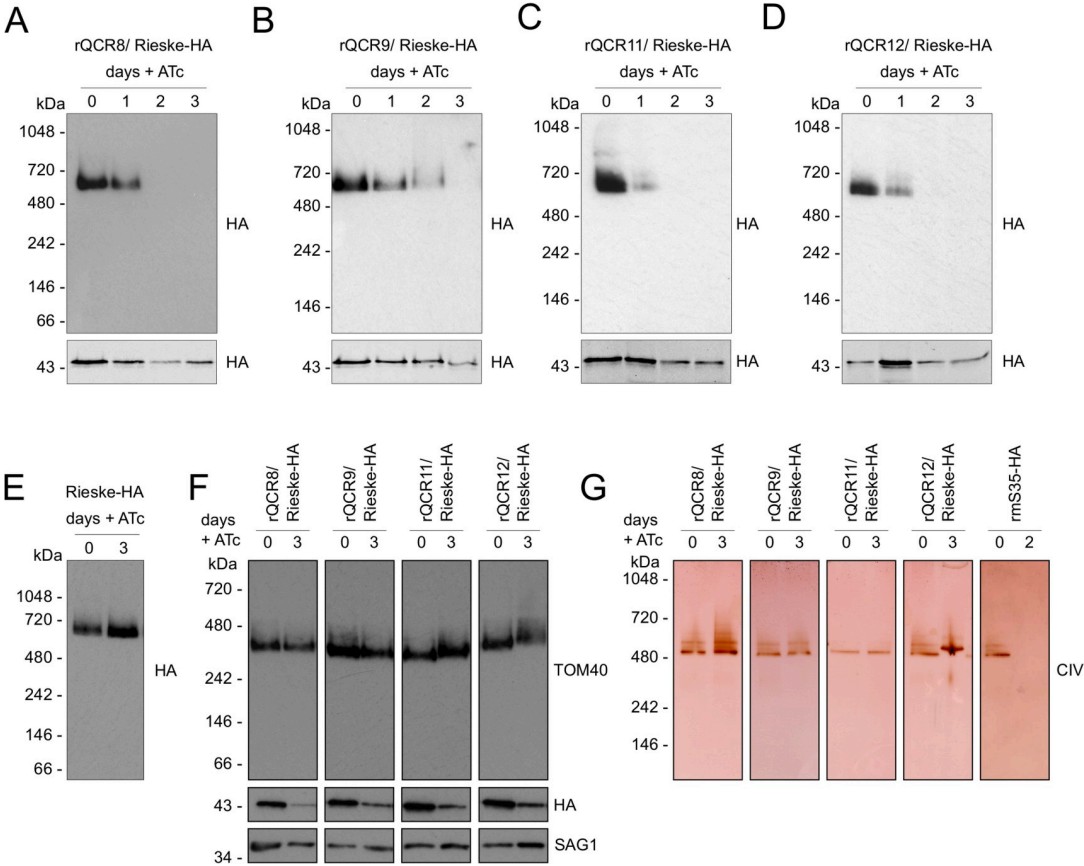

**Fig 6. Loss of each of the four putative subunits affects stability of complex III. (A-D)** Total lysate from parasites from promoter replacement lines in the Rieske-HA background (rQCR8/9/11/12 / Rieske-HA) grown in the absence (0) or presence of ATc for 1–3, were separated by BN-PAGE, blotted and immunolabelled with antibodies against HA. Samples were also separated by SDS-PAGE and immunolabelled with anti-HA antibodies. **(E)** Total lysate from the parental Rieske-HA line grown in the absence (0) or presence of ATc for 3 days were separated by BN-PAGE, blotted and immunolabelled with anti-HA antibodies. **(F)** Total lysate from promoter replacement lines in the Rieske-HA background (rQCR8/9/11/12 / Rieske-HA) grown in the absence (0) or presence of ATc for 3 days (3) were separated by BN-PAGE and immunolabelled with anti-TOM40. Samples were also separated by SDS-PAGE and immunolabelled with anti-HA and anti-SAG1 antibodies. **(G)** Total lysate from promoter replacement lines in the Rieske-HA background (rQCR8/9/11/12 / Rieske-HA) grown in the absence (0) or presence of ATc for 3 days (3) were separated by clear native-PAGE and complex IV oxidation activity was performed. A line where ribosomal protein mS35 is downregulated was used as positive control for loss of complex IV [34].

Using complexome profiling, we identified 27 of the 29 previously discovered subunits of ATP synthase, including 13 of the recently described apicomplexan-unique subunits [6, 18]. One of the subunits we were unable to detect (TGGT1_249720, subunit *c*), has proven elusive by mass spectrometry, also being absent from other *Toxoplasma* mitochondrial proteome datasets [6, 7, 18, 36]. The migration of ATP synthase at ~1860 kDa (**S5 Table**) is higher to that observed in previous studies, which may be due to the different detergent conditions used (DDM in this study vs digitonin [6]), but which is consistent with the migration observed using an anti-beta subunit antibody (**S4A Table**). The ATP synthase complex of *Toxoplasma* is therefore likely to be larger than the ~1250 kDa dimer [43] observed in yeast, probably due to the addition of many lineage-specific subunits. Our complexome identified three putative new subunits of ATP synthase which have not previously been detected. We name these ATPTG1, ATPTG7 and ATPTG16 in line with a recent structure of *Toxoplasma* ATP synthase that assigned new nomenclature and that observed these three proteins as part of the structure

[41]. Interestingly, TGGT1_246540 appears to encodes both the ATP synthase subunit ATPTG1 and the complex III subunit CytC1. One possibility is that TGGT1_246540 encodes two separate proteins. Another is that, like its yeast homolog [65] the protein is post-translationally cleaved upon mitochondrial import. An example where the N-terminal targeting sequence is retained after cleavage is the case of the mammalian Rieske protein, for which the cleaved N-terminus is retained as part of complex III [66]. Several observations support a model of post-translational cleavage: the C-terminally HA tagged CytC1 protein migrates at ~35 kDa, 10 kDa lower than the ~45 kDa predicted from the gene model. Likewise, in eight IP experiments for complex III that we performed (five using CytC1, and two using Rieske, as baits) only C-terminal peptides of TGGT1_246540, from the portion of the protein predicted to encode the complex III subunit CytC1, are identified (**S11 Table**). Finally, the atomic model of ATP synthase resolved only the N-terminal 153aa of ATPTG1 as part of the structure [41]. The ATPTG1 subunit appears myzozoan-specific but is only fused to a cytochrome $c_1$ domain in a phylogenetically restricted group of apicomplexans. The evolution of this subunit deserves future study.

Our complexome dataset predicts a size of ~460 kDa for complex IV (**S5 Table**). This is smaller than previous estimates (for example the ~600 kDa suggested by Seidi et al [7]), which may be due to different detergent conditions or differences in mass estimation accuracy. Three subunits assigned apicomplexan specific (ApiCox) by Seidi et al [7] have similarity to mammalian complex IV subunits. As noted by Seidi et al, ApiCox23 corresponds, according to HHPRED [57], to cytochrome *c* oxidase complex subunit 4 and ApiCox25 to subunit 6A. We propose renaming these subunits as TgCox4 and TgCox6a. In addition, ApiCox 26 (TGGT1_306670) has similarity to mammalian cytochrome *c* oxidase complex subunit NDUFA4 (92.64% probability, e-value 0.35, between amino acids 69–114, PDB annotation: 5Z62_N) and so should be renamed TgNDUFA4. Our analysis detected six putative new subunits co-migrating with the complex (**Fig 2D**). All six were detected previously, but not recognised as complex IV subunits [7, 34]. Five of these proteins are restricted to the apicomplexans, while one, TGGT1_200310, appears myzozoan specific (**S4 Fig**). The existence of these evolutionary divergent subunits underlines the highly divergent nature of the apicomplexan/myzozoan cytochrome *c* oxidase complex. It remains to be investigated whether the divergent *Toxoplasma* complex can perform parasite-specific functions and what the role of the recently discovered subunits are.

Compared to well-studied organisms, little is known about complex II (succinate dehydrogenase) in apicomplexan parasites. Clear homologs of mammalian SDHA and SDHB can readily be identified by BLAST searches [14, 17, 27]. SDHA contains a flavin adenine dinucleotide (FAD) cofactor and SDHB contains three Fe-S clusters. These subunits transfer electrons from succinate to ubiquinone, reducing it to ubiquinol for reoxidation by complex III. Unexpectedly, our analysis was unable to resolve the SDHA homolog (TGGT1_215590) in the same gel slice as other complex II subunits, raising a question about its association with complex II. However, TGGT1_215590, is suggested to be mitochondrial in both the LOPIT [36] and mitochondrial matrix proteome [7], and has a low phenotype score [56], all of which are consistent with its assignment as a complex II subunit. Future experiments, for example immunoprecipitation, will be required to confirm if SDHA associates with other complex II subunits. The SDHB subunit migrated at ~500 kDa in our complexome (**Fig 2D**), significantly larger than the ~130 kDa complex seen in mammals and yeast [37, 43]. Native PAGE and immunoblotting of the triple-HA epitope tagged SDHB subunit confirmed this finding (**Fig 3C**), which is also in line with a recent study of the apicomplexan *Eimeria tenella* [46]. The mammalian complex contains two other subunits, SDHC and SDHD, which function as hydrophobic membrane anchors. Both subunits together bind a single haem group, and SDHD also has a ubiquinone

binding site. No clear homologs of SDHC or SDHD have been identified in apicomplexans. Two proteins were suggested to correspond to SDHC/D in *Plasmodium* [67, 68], but have yet to be confirmed. The suggested *Pf*SDH4 (PF3D7_1010300) does not have a clear homolog in *Toxoplasma*. TGGT1_320470 appears to be a homolog for the proposed *Pf*SDH3 (PF3D7_0611100), however, while TGGT1_320470 is present in our mass spectrometry database, we did not detect it in our complexome analysis and its mitochondrial localisation is not supported by previous proteomic approaches [7]. Additionally, a recent preprint study investigating mitochondrial complexes in *P. falciparum* found that the proteins encoded by PF3D7_1010300 and PF3D7_0611100 did not co-migrate with other complex II subunits [69]. Together these data suggest that these two proteins are probably not complex II subunits.

Here, we identified seven proteins which co-migrated with the highly conserved SDHB subunit. Five were suggested to be mitochondrial and four had a similar phylogenetic distribution in myzozoans as SDHA and SDHB, supporting their candidacy as novel lineage-specific complex II subunits. If confirmed, this would represent the largest number of subunits for a eukaryotic complex II, and explain the large observed size of the complex in apicomplexans. The sum of the masses of known and novel complex II subunits is ~190 kDa (**S8 Table**), suggesting either that more subunits await discovery or that the complex migrates in native PAGE as a higher order oligomer.

Apicomplexan complex III is a clinically validated drug target, and numerous inhibitors are in therapeutic use. Hydroxynaphthoquinones, including atovaquone and buparvaquone, and decoquinate are thought to bind to the $Q_o$ site, while others, including endochin-like quinolones and 4-pyridones, are thought to target the $Q_i$ site [53–55, 70–73]. However, despite widespread clinical use, the mechanisms of drug action are not well understood, due to lack of knowledge of the complex's composition and structure. The increased susceptibility of apicomplexan complex III to these inhibitors compared to other organisms suggests divergent features. Previously, six well-conserved complex III subunits have been identified using bioinformatic approaches [14, 17, 27], although none have been experimentally studied, with the exception of QCR2/TGGT1_202680 which was localised in a previous study [48]. Here we confirm that these subunits co-migrate in a ~670 kDa complex by BN-PAGE and localise to the mitochondrion. Beyond these conserved subunits, we investigated the novel subunits QCR8, QCR9, QCR11 and QCR12. Downregulation of each of the four genes resulted in a severe growth defect and resulted in a specific defect in complex III stability. Downregulation of all four of these genes resulted in disruption of the mitochondrial membrane potential, consistent with a proposed role for these proteins in stability of the complex. Recent structural studies of *S. cerevisae* respiratory supercomplexes suggest that ScQCR9 and ScQCR10 bind to the transmembrane region of Rip1 protein (the Rieske subunit) [26, 74], thereby securing it on the complex. *Toxoplasma* QCR9 is the homolog of ScQCR9 and may play a similar role. ScQCR10 does not have a *Toxoplasma* homolog and so it is possible that TgQCR11 and or TgQCR12 play a similar structural role. Overall, we identified 11 *Toxoplasma* subunits of complex III which combine to a predicted weight of 305 kDa (**S8 Table**). We observed a complex of approximately twice this size, ~670 kDa, which matches its expected dimeric nature, suggesting that we identified the majority of *Toxoplasma* complex III subunits.

In other eukaryotes, mETC complexes often associate together to form respiratory supercomplexes [43, 58, 59]. Their exact physiological role is still debated, but they have been proposed to play a role in decreasing reactive oxygen species production, assisting in complex assembly, regulating mETC activity, and preventing aggregation in the inner membrane [58]. The presence of supercomplexes in *Toxoplasma* has been hypothesised [7] but never observed. Here, using mild non-ionic detergents, we observe the Rieske, CytC1 and QCR2 subunit in a complex of higher molecular weight than the complex III dimer, providing the first evidence

of a supercomplex formation in Apicomplexa. In mammals, complex III can associate with complex I, complex IV or both complex I and IV to form several supercomplexes [58]. Given the absence of complex I in apicomplexans, *Toxoplasma* complex III may simply associate with complex IV, like in yeast and mycobacterium [26, 74–76]. It is also possible, given the divergence of the apicomplexan mETC, that novel parasite-specific respiratory supercomplexes are formed which could have relevance in control of metabolism.

The mitochondrial genome of *Toxoplasma* has recently been sequenced [42]. It contains three protein encoding genes, which encode three mETC subunits: CoxI, CoxIII and CytB. Prior to this, the exact gene models and sequences were uncertain, hindering efforts to identify and correctly assign peptides in proteomic studies. A recent study of the mitochondrial matrix proteome [7] did not identify any of the three proteins, while the whole cell LOPIT proteome could only identify CytB [36]. Using the recent sequence data, we were able to identify all three proteins and found that they co-migrate with other subunits from complex III (CytB) and complex IV (CoxI, CoxIII).

In this study, we have elucidated the composition of apicomplexan complex III. An independent study has also looked at *Toxoplasma* complex III [29] and identified the same 11 subunits. As these studies were carried out in parallel and using different methods, these findings provide independent support of each other. Understanding the subunit composition of the mETC and $F_1F_o$-ATP synthase is the first step to understanding how respiration occurs and is regulated in these obligate parasites. Any divergence in these essential processes from their mammalian hosts might present targets for future drug discovery.

## Materials and methods

### Cell culture and growth analysis

*T. gondii* tachyzoites were cultured in human foreskin fibroblasts (HFF), sourced from ATCC, catalogue number #CRC1041, or in Vero cells, derived from *Chlorocebus sabaeus* kidney epithelial cells, for mitochondrial isolation experiments. HFF and Vero cells were cultured in Dulbecco's Modified Eagle's Medium (DMEM), supplemented with 10% (v/v) fetal bovine serum, 4 mM L-glutamine and Penicillin/Streptomycin antibiotics and grown at 37˚C with 5% $CO_2$. Where relevant anhydrotetracycline (ATc) was added to the growth medium at a final concentration of 0.5 μM. For plaque assays fresh HFFs were infected with parasites in the presence or absence of ATc for 9–10 days. Cells were fixed with methanol and stained with crystal violet staining solution.

### Plasmid construction and stable transfection

C-terminal triple HA epitope tagging: gRNAs targeting the stop codon of each GOI were identified with the ChopChop tool (https://chopchop.cbu.uib.no/) and were cloned into a U6 promoter and CAS9-GFP expressing vector (Tub-Cas9YFP-pU6-ccdB-tracrRNA) [77] using the BsaI restriction site. The CAT selection cassette and triple HA epitope were amplified by PCR from a p3HA.LIC.CATΔpac plasmid [47, 78]. The gRNA/CAS9 vector-PCR product mixture was transfected into the TATiΔ*ku80* [47] line by electroporation and cassette integration was selected with chloramphenicol. Positive clones were isolated by serial dilution and confirmed by PCR analysis.

Promoter replacement: gRNAs targeting the start codon of each GOI were identified and cloned into the Cas9-GFP expressing vector as detailed above. The DHFR selectable cassette and ATc repressible promoter were amplified by PCR from pDT7S4myc [47, 79]. The gRNA/

CAS9 vector-PCR product mixture was transfected into a line containing an HA-tagged version of the Rieske subunit line by electroporation and cassette integration was selected with pyrimethamine. Positive clones were isolated by serial dilution and confirmed by PCR analysis.

## qRT-PCR

In each experiment, parasites were cultured in triplicate in the absence or presence ATc for 2 days. Cells were harvested and large host cell debris were removed by a 3 mircon filter. RNA was extracted using RNeasy kit (Qiagen) with an added DNAseI step. cDNA was made with High capacity RNA-to-cDNA kit (AppliedBiosystems). qRT-PCR was set up with PowerSYBR green master mix (ThermoFisher) using 10 ng of cDNA as a template. Water only and RNA (10 ng) were included as controls. qRT-PCR was run with 7500 Real Time PCR System (Applied Biosystems) using default temperature settings and performing a dissociation step after each run. Expression in +ATc samples relative to -ATc samples was calculated using the double $\Delta$ Ct method [80] using catalase (TGGT1_232250) as internal control. Data was plotted in GraphPad Prism 8.4.3 and treatments compared with an unpaired *t*-test.

## Blue and clear-native PAGE

For blue-native PAGE, whole parasite or mitochondrial protein samples were suspended in solubilisation buffer (750 mM aminocaproic acid, 50 mM Bis-Tris–HCl pH 7.0, 0.5 mM EDTA, 1% (w/v) detergent: dodecyl maltoside, digitonin or OGP) and incubated on ice for 15 minutes. The mixture was centrifuged at $16,000 \times g$ at 4˚C for 30 minutes. The supernatant containing solubilised membrane proteins was combined with sample buffer containing Coomassie G250 (NativePAGE), resulting in a final concentration of 0.25% DDM and 0.0625% Coomassie G250. Samples (equivalent of 2.5 or 5 x $10^6$ parasites per lane for immunoblotting) were separated on a on NativePAGE 4–16% Bis-Tris gel. NativeMark was used as a molecular weight marker.

Clear-native PAGE and complex IV activity stain was performed as described previously [34]. Briefly, whole parasite samples were suspended in solubilisation buffer (50 mM NaCl, 50 mM Imidazole, 2 mM 6-aminohexanoic acid, 1 mM EDTA–HCl pH 7.0, 2% (w/v) n-dodecyl-maltoside) and incubated on ice for 10 minutes. The mixture was centrifuged at $16,000 \times g$ at 4˚C for 15 minutes. The supernatant containing solubilised membrane proteins was combined with glycerol and ponceau S to a final concentration of 6.25% and 0.125% respectively. Samples (equivalent of 1.5 x $10^7$ parasites per lane) were separated on a on NativePAGE 4–16% Bis-Tris gel. NativeMark was used as a molecular weight marker. Complex IV oxidation activity was shown by incubating gels in a 50 mM $KH_2PO_4$, pH 7.2, 1 mg ml−1 cytochrome *c*, 0.1% (w/v) 3,3′-diaminobenzidine tetrahydrochloride solution.

## SDS-PAGE and immunoblot analysis

Total cell samples (2.5 x $10^6$ parasites per lane) were resuspended in Laemmli buffer (2% (w/v) SDS, 125 mm Tris–HCl pH 6.8, 10% (w/v) glycerol, 0.04% (v/v) β-mercaptoethanol, and 0.002% (w/v) bromophenol blue) and boiled at 95˚C for 5 minutes. Proteins were then separated on a 12.5% SDS-PAGE gel. EZ-Run Prestained Rec protein ladder was used as a molecular weight marker. Proteins were transferred under semi-dry conditions to nitrocellulose membrane (0.45 μm Protran), labelled with the appropriate primary antibodies: anti-HA (1:500, Sigma), anti-ALD (1:2000), anti-Mys/TGME49_215430 ([81], 1:2000), anti-TOM40 ([48], 1:2000), anti-ATPβ (Agrisera AS05085, 1:5000), IMC1 ([82], 1:1000), CPN60 ([83], 1:2000), SAG1 (gift from the Soldati lab, 1:2000), and detected using either secondary

horseradish peroxidase-conjugated antibodies and chemiluminescence detection using Pierce ECL Western Blotting Substrate and an x-ray film or using secondary fluorescent antibodies IRDye 800CW, 680RD (1:10000, LIC-COR) and detection with an Odyssey CLx.

For the blue native-PAGE, proteins were transferred onto a PVDF membrane (0.45 μm, Hybond) using wet transfer in Towbin buffer (0.025 M TRIS 0.192 M Glycine 10% Methanol) for 60 minutes at 100 V. After transfer, immunolabelling and visualisation was carried out as described above by chemiluminescence detection.

### Immunofluorescence assay and microscopy

Parasites were inoculated on fresh HFFs on glass coverslips. After 1-day cells were fixed with 4% paraformaldehyde. Cells were permeabilised and blocked with a solution of 2% bovine serum albumin and 0.2% triton X-100 in PBS before incubation with primary antibodies (anti-HA, Sigma, 1:1000 and rabbit anti-TOM40, [48], 1:1000, followed by secondary antibodies (Alexa Fluor Goat anti-Rat 488 Invitrogen #A-11006, 1:1000 and Alexa Fluor Goat anti-Rabbit 594 Invitrogen #A-11012, 1:1000). Coverslips were mounted on slides with Fluoromount-G mounting media containing DAPI (Southern Biotech, 0100–20). Images were acquired on a DeltaVision Core microscope (Applied Precision) using the 100x objective as previously described [84]. Images were processed and deconvolved using the SoftWoRx and FIJI software.

To assess mitochondrial morphology parasites were grown in the presence or absence of 0.5 μM ATc for 2 days. Mitochondria were visualised by immunofluorescence, as described above, using anti-Mys/TGME49_215430 ([81], 1:1000). 150–200 parasites of each treatment were counted, and mitochondrial morphology was assessed as normal or abnormal as described previously [34]. Experiments were performed in triplicate.

### Mitochondrial isolation

Parasite tachyzoites from a line with a mitochondrial ribosomal subunit HA tagged (rTgmS35; [34]) were grown in Vero cells and harvested (1.5 x $10^{10}$ parasites). Parasite cells were lysed by nitrogen cavitation and a mitochondrially enriched pellet isolated by differential centrifugation as previously described [34]. This crude mitochondrial pellet was further purified on a discontinuous Percoll gradient (buffer composition 18/23/40% Percoll) by centrifugation (40,000 x g, 1hr, 4˚C) in an SW41 rotor (Beckman Coulter) and enriched mitochondria were collected from the 23–40% interface (adapted from [85]). Fractions at each interface were also taken, pelleted and protein content determined by a Bradford assay. Fractions were analysed by immunoblot to determine their composition.

### Complexome profiling

Blue native PAGE/complexome profiling was performed as described previously [86], based on the method developed in [30]. Briefly, 20 μg of mitochondrial sample was separated by BN-PAGE as described above, with a DDM: protein ratio of 2.5:1. The gel lane was cut into 63 equally sized slices before being digested with trypsin and peptides extracted by the addition of 60% acetonitrile and 4% formic acid. A portion of each extraction (1–3 ul) was added straight into 10 ul of 0.1% TFA, 3% acetonitrile and 10 μl was used for analysis. Samples were fractionated on an Acclaim PepMap nanoViper C18 reverse-phase column (Thermo Scientific) (75 μm × 150 mm), with a gradient of 5–40% acetonitrile in 0.1% formic acid at a flow rate of 300 nL min$^{-1}$ over 84 min, then peptides were analysed by a Q-Exactive Plus Orbitrap mass spectrometer (Thermo Scientific) with fragmentation performed by higher-energy collisional dissociation (HCD) using nitrogen. The mass range was 400 to 1600 m/z for the precursor

ions; the top 10 most abundant ions were selected for MS/MS analyses. Spectra were assigned to peptide sequences and originating proteins using Mascot 2.4 (Matrix Science Ltd.) with a peptide precursor mass tolerance of 5 ppm and fragment mass tolerance of 0.01 Da, allowing for up to two missed cleavages and variable modifications (methionine oxidation, cysteine propionamide, protein N-formylation and N-acetylation). Only peptide assignments made by Proteome Discoverer scoring at or above its P < 0.05 threshold versus a decoy database were considered. 67 out of 76 proteins of interest (those displayed in **Figs 2D and S7**) were identified by 2 or more peptides. 7 of the 9 proteins identified by 1 peptide scored at or above a P < 0.01 threshold. The 2 remaining proteins, TGGT1_257160 and CytB, scored at or above its P < 0.05 threshold but not the P < 0.01 threshold.

To assign peptide sequences the Proteome Discoverer (Thermo Fisher Scientific) software was used with Mascot 2.4 (Matrix Science Ltd.) configured to use a custom database containing *Toxoplasma gondii* GT1 proteins (ensembl proteins toxoplasma_gondii_gt1_gca_00149715.TGGT1.pep.all.fa last updated 04.03.2020) in addition to *Chlorocebus sabaeus* ensembl proteins (Chlorocebus_sabaeus.ChlSab1.1.pep.all last updated 05.03.2020) with the manual addition of the protein sequences for *T. gondii* mitochondrial encoded proteins CoxI, CoxIII and CytB [42], plus NCBI reference sequence XP_007959728. A separate copy of the database was further modified to truncate sequences for $F_1F_o$ ATP synthase subunits alpha and beta and succinate dehydrogenase subunit A to before a peptide that is common between the species, to avoid erroneous apparent co-migration.

Protein quantification was performed per slice using Proteome Discoverer. Abundances for all well-assigned peptides observed in all the samples was counted and combined into a multi-consensus spreadsheet using default parameters. Its standard aggregation algorithm was used to measure the relative protein abundance for each protein in each gel slice by taking the quantified precursor peak area information and normalising to the highest point for each protein, which was set to 1. Clustering was performed using Pearson distribution and Heatmaps were generated using Nova v0.8.0 [87]. Full datasets used for complexome profiling were deposited in the ComplexomE profiling DAta Resource (CEDAR) database (https://www3.cmbi.umcn.nl/cedar/browse/experiments/CRX27).

Proteins identified in our analysis were compared to existing proteomic datasets [7, 36] using ToxoDB.org (release 46, accessed August 2020).

## JC-1

Parasites grown in the presence or absence of 0.5 μM ATc for 2 or 3 days, as indicated. Intracellular parasites were lysed out of host cells using a 26G needle filtered through a 3 μm filter and incubated in their growth media with 1.5 μM of JC-1 (5,5',6,6'-Tetrachloro-1,1',3,3'-tetraethylbenzimidazolocarbocyanine iodide, Thermo Fisher Scientific, stock 1.5 mM in DMSO) for 15 minutes at 37°C before analysis using BD FACSCelesta (BD Biosciences). Treatment with 10 μM valinomycin for 30 minutes was included prior to JC-1 incubation as a depolarising control. Treatment with 1μM atovaquone and 200 μM antimycin a for 4 hours was included prior to JC-1 incubation as a complex III inhibitory control. Unstained controls were used to define gates for analysis. 100,000 events per treatment, were collected and data were analysed using the FlowJo software to define the population of parasites with red fluorescent signal.

## Immunoprecipitations

Parasites (Rieske-HA or CytC1-HA) were grown (1–2 x10^8 cells) on HFFs cells and harvested. Parasites were lysed with a 1% DDM or triton X-100 solution and immunoprecipitated with

Pierce anti-HA agarose beads. Bound proteins were eluted with 50 mM NaOH. Eluted proteins were detected by mass spectrometry as described previously [34].

## Supporting information

**S1 Fig. Mitochondrial isolation and complexome profiling workflow.** Schematic representation of the workflow for complexome profiling: a large number of *Toxoplasma* tachyzoites (~1 x $10^{10}$) were harvested and then cells lsyed using nitrogen cavitation. Three different centrifugation conditions were used to obtain a mitochondrially enriched fraction: differential centrifugation (at 1,500 x g and 16,000 x g) was used to separate organelles from unbroken cells and heavy cell debris. This fraction was then layered onto a Percoll step gradient and then, after density-gradient fractionation at 40,000 x g, recovered from the 23%/40% Percoll interface. Proteins were then separated on a Blue Native PAGE gel and the gel cut into 61 gels slices. Each gel slice was subjected to mass spectrometry and label free quantification performed. The resulting data allowed calculation of the relative abundance of individual proteins in each slice, and this is visualised using a heatmap. The total dataset gives the relative abundance information for hundreds of proteins across all gel slices.
(TIF)

**S2 Fig. Mass calibration of complexes and subunit profiles. (A)** Mitochondrial samples separated by blue native PAGE used to generate the complexome profile. The gel was cut into 61 gel slices as indicated on the right of the gel and molecular weights, based on the masses of mammalian complexes, indicated to the right of the gel slice numbers. The identities of host cell and *Toxoplasma* complex Coomassie-stained bands are indicated to the left of the gel strip. **(B)** Graphs depicting complexome profiles of *T. gondii* ATP synthase (V, red), complex IV, III and II (yellow, blue, green respectively) through a representative subunit from each complex (gamma for ATP synthase, Cox2a, Rieske and SDHB for complexes IV, III, II respectively). The x-axis depicts the gel slice number and the y-axis the protein's relative abundance. The full complexome profiling dataset is provided in **S1 Table. (C)** Graphs depicting complexome profiles of all *T. gondii* subunits from ATP synthase (complex V), complex IV, III and II. The x-axis depicts the gel slice number and the y-axis the protein's relative abundance. The full complexome profiling dataset is provided in **S1 Table. (D)** Mass calibration using *C. sabaeus* host cell complexes of known or predicted size. The x-axis depicts the molecular mass in kDa and the y-axis the gel slice number. Full detail of mass calibration is provided in **S5 Table**. AK2—Adenylate kinase 2; LAS—Lipoic acid synthetase; SQOR—Sulfide:quinone oxidoreductase; ETF—Electron-transferring flavoprotein A+B; ETFDH—ETF dehydrogenase; IDH—Isocitrate dehydrogenase; CS—Citrate synthase; Cx II—Complex II; LDH—Lactate dehydrogenase complex; Cx IV—Complex IV; Fum—Fumarase; GDH—Glutamate dehydrogenase; Cx III—complex III; Cx III2 +IV—Complex III + IV; Cx V—ATP synthase; Cx I—Complex I.
(TIF)

**S3 Fig. Profiles of subunits with common peptides between *Toxoplasma* and *Chlorocebus*.** Graphs depicting complexome profiles and alignments of ATP synthase subunit alpha **(A)** and ATP synthase subunit beta **(B)** from *T. gondii* (red, Tgα and Tgβ) and *C. sabaeus* (black, Csα and Csβ) before and after truncation of common peptides. ATP synthase subunit gamma from *T. gondii* (grey, dashed line, Tgγ) depicts the position of *T. gondii* ATP synthase subunits. The x-axis depicts the gel slice number and the y-axis the protein's relative abundance. Common peptides between the two proteins are outlined in the alignment by a red box, and *Toxoplasma* specific peptides after truncation are marked by a blue box. Alignments were performed by

Clustal Omega. Full detail of protein truncation is provided in **S6 Table**.
(TIF)

**S4 Fig. Analysis of putative *Toxoplasma* ATP synthase subunits. (A)** Total lysate from parasite cells separated by BN-PAGE and immunolabelled with antibodies against the betta subunit of ATP synthase (ATPβ). **(B)** Table showing examples for two previously identified subunits—ATP synthase subunit beta and ATP synthase subunit gamma (ATPβ,γ) (known) and complexome identified putative novel ATP synthase subunits—ATPTG1,7,16 (novel) and their homology distribution across key groups. Homology searches were performed using the HMMER tool [57]. Coloured circles refer to the e-value from the HMMER search: white indicates a hit with an e-value above 0.0001, black indicates no hits, and red indicates hits with an e-value below 0.0001, as indicated in the coloured scale. Full data are given in **S7 Table**. *Hh*: *Hammondia hammondi*; *Pb*: *Plasmodium berghei*; *Cp*: *Cryptosporidium parvun*; *Cyryptosporidium muris*; *Vb*: *Vitrella brassicaformis*; *Pm*: *Perkinsus marinus*; *Sm*: *Symbiodinium microadriaticum*; *Pt*: *Paramecium tetraurelia*. **(C)** Graph depicting peptide abundances detected for TGGT1_246540 in the complexome profile and the amino acid sequence of the protein with the peptides detected in colour. Peptides from the N-terminal portion of the protein, which constitute ATPTG1 are marked in cyan, and peptides in the C-terminal portion of the protein, which constitute CytC1, are marked in magenta. **(D)** Schematic diagram of TGGT1_246540 showing the amino acids that make up ATPTG1 (cyan) and the cytochrome c1 domain (magenta). Residues in bold show the detected peptides.
(TIF)

**S5 Fig. Analysis of putative new *Toxoplasma* putative complex IV subunits.** Table showing previous predicted (known) and complexome identified putative novel (novel) complex II subunits and their homology distribution across key groups. Homology searches were performed using the HMMER tool [57]. Coloured circles refer to the e-value from the HMMER search: white indicates a hit with an e-value above 0.0001, black indicates no hits, and red indicates hits with an e-value below 0.0001, as indicated in the coloured scale. Full data are given in **S7 Table**. *Hh*: *Hammondia hammondi*; *Bb*: *Babesia bovis Pf*: *Plasmodium falciparum; Pb*: *Plasmodium berghei*; *Cp*: *Cryptosporidium parvun*; *Cyryptosporidium muris*; *Vb*: *Vitrella brassicaformis*; *Pm*: *Perkinsus marinus*; *Sm*: *Symbiodinium microadriaticum*; *Pt*: *Paramecium tetraurelia*.
(TIF)

**S6 Fig. Generation of endogenous HA tagged lines. (A)** Schematic depiction of the endogenous HA-tagging strategy of a gene of interest (GOI). (i) CRISPR/CAS9 guided cut at the predicted GOI/UTR boundary, (ii) a repair cassette containing the triple hemagglutinin epitope tag (3HA), the Chloramphenicol acetyltransferase (CAT) selection marker, and homology to the GOI/UTR boundary, is inserted between the GOI and UTR during cut repair guided by homology regions, (iii) GOI with the integrated repair cassette. The black arrows represent the four primers used to confirm integration via the PCRs shown in **B**. **(B)** Confirmation of generation of 3HA-tagging at the desired loci via PCR analysis using primers 1–4 (primers in **S12 Table**) shown in **A**.
(TIF)

**S7 Fig. Complexome heatmap of additional mETC dehydrogenases.** Heatmaps of the complexome profiles of additional *Toxoplasma* mitochondrial mETC dehydrogenases. Heatmaps represent the 61 gel slices the BN-PAGE gel was cut into, from the top (left) to the bottom (right) of the gel. Molecular weight markers shown on the top are based the migrations of mammalian mitochondrial complexes of known size. Protein names are shown on the left of their respective profiles. Red indicates the highest relative abundance (1) and black the lowest

(0).
(TIF)

**S8 Fig. Generation of promoter replacement strains of four novel complex III subunits.**
**(A)** Schematic depiction of the promoter replacement strategy allowing knock-down of a gene of interest (GOI) with the addition of anhydrotetracycline (ATc). (i) CRISPR/CAS9 guided cut at the predicted promoter/ATG boundary, (ii) a repair cassette containing the ATc repressible promoter, the dihydrofolate reductase (DHFR) selection marker, and homology to the promoter/ATG boundary, is inserted between the promoter and GOI during cut-repair guided by the homology sequences, (iii) GOI, under the control of ATc repressible promoter, is down regulated when ATc is added. The black arrows represent the four primers used to confirm integration via the PCRs shown in **B** (primers in **S12 Table**). **(B)** Confirmation of generation of promoter replacement at the desired loci via PCR analysis using primers 1–4 (primers in S9 Table) shown in **A**. **(C)** Transcript levels of each gene (QCR8,9,11,12) were analysed by qRT-PCR, in the absence (-) or presence (+) of ATc after 2 days. Bars represent the mean ± SEM (n = 3).
(TIF)

**S1 Table. Complexome data—Calculated relative abundance for each protein identified in the Complexome analysis.** A) Complexome profiles of *Toxoplasma* proteins displayed in Figs 2D and S7. B) Complexome profiles of all *Toxoplasma* proteins.
(XLSX)

**S2 Table. Proteomics summary data–score, predicted protein coverage and number of unique peptides found for each *Toxoplasma* protein found in the complexome analysis and *Chlorocebus* proteins used for mass calibration.** A) *Toxoplasma* proteins displayed in Figs 2D and S7. B) *Chlorocebus* proteins used for mass calibration. C) Complete list of *Toxoplasma* proteins detected.
(XLSX)

**S3 Table. Proteome comparisons–analysis of presence of complexome identified proteins in previous proteomics studies with mitochondrial proteomes.**
(XLSX)

**S4 Table. Mitoribosomal proteins found in the complexome dataset.**
(XLSX)

**S5 Table. Mass calibration–calculation of apparent mass found in each gel slice.**
(XLSX)

**S6 Table. Truncated proteins–abundance profiles of proteins before and after truncation of peptides that are mutual to the *Toxoplasma* and *Chlorocebus* homologs of ATP synthase and complex II subunits.**
(XLSX)

**S7 Table. HMMER homology–summary of data obtained from homology searches performed via HMMER for the complexome identified *Toxoplasma* subunits of each complex.**
(XLSX)

**S8 Table. Subunit masses–prediction of mass of each complexome identified *Toxoplasma* subunits while considering predicted mitochondrial targeting sequence removal.**
(XLSX)

**S9 Table. mETC and ATP synthase subunits–summary information of all known and predicted *Toxoplasma* mETC and ATP synthase subunits.**
(XLSX)

**S10 Table. Mitochondrial morphology–counts of normal and abnormal mitochondrial morphology in rQCR8,9,11,12/ Rieske-HA lines in the presence and absence of ATc.**
(XLSX)

**S11 Table. TGGT1_246540 immunoprecipitation–peptides of TGGT1_246540 detected in immunoprecipitation experiments.**
(XLSX)

**S12 Table. Primers–summary of primers used in this study for generation and confirmation of gene tagging and promoter replacement.**
(XLSX)

## Acknowledgments

We thank Leandro Lemgruber from the imaging facilities of the Wellcome Centre for Integrative Parasitology, Diane Vaughn and Alana Hamilton from the cellular analysis facility of the Institute of Infection, Immunity and Inflammation at the University of Glasgow and Michael Harbour and Ian Fearnley (MBU mass spectrometry facility) for technical support. We thank Prof. Ron Dzikowski for providing experimental facilities at the Kuvin Centre, Hebrew University of Jerusalem during the Covid19 lockdown.

## Author Contributions

**Conceptualization:** Andrew E. Maclean, Lilach Sheiner.

**Data curation:** Andrew E. Maclean, Hannah R. Bridges, Judy Hirst, Lilach Sheiner.

**Formal analysis:** Shujing Ding, Jana Ovciarikova.

**Funding acquisition:** Andrew E. Maclean, Judy Hirst, Lilach Sheiner.

**Investigation:** Andrew E. Maclean, Hannah R. Bridges, Mariana F. Silva, Shujing Ding, Jana Ovciarikova.

**Methodology:** Andrew E. Maclean, Hannah R. Bridges, Judy Hirst, Lilach Sheiner.

**Supervision:** Judy Hirst, Lilach Sheiner.

**Writing – original draft:** Andrew E. Maclean, Judy Hirst, Lilach Sheiner.

**Writing – review & editing:** Andrew E. Maclean, Hannah R. Bridges.

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
