## [Decision Letter · Decision Letter 0]

21 Sep 2020

Dear Lilach,

Thank you very much for submitting your manuscript "Complexome profile of Toxoplasma gondii mitochondria identifies a divergent cytochrome bc1 complex" for consideration at PLOS Pathogens. As with all papers reviewed by the journal, your manuscript was reviewed by members of the editorial board and by several independent reviewers. In light of the reviews (below this email), we would like to invite the resubmission of a significantly-revised version that takes into account the reviewers' comments.

As you will see, all 3 reviewers were highly enthusiastic about this work and agreed on its importance to the field. However, some concerns were raised that require in some cases additional experimental work and clarification. In particular, the fate of the leader sequence of the cyt c1 protein requires further investigation, the question of whether atovaquone depolarizes the Toxoplasma mitochondrion is raised, and further controls are requested for loss of complex III signal upon knock-down of QCR proteins 8-12. It is also requested that the set of mitoribosomal proteins identified in the proteomic analysis is described in a stand-alone table, and a number of additional issues are raised that require attention.

We cannot make any decision about publication until we have seen the revised manuscript and your response to the reviewers' comments. Your revised manuscript is also likely to be sent to reviewers for further evaluation.

Sincerely,

Michael J Blackman

Associate Editor

PLOS Pathogens

Vern Carruthers

Section Editor

PLOS Pathogens

Kasturi Haldar

Editor-in-Chief

PLOS Pathogens

orcid.org/0000-0001-5065-158X

Michael Malim

Editor-in-Chief

PLOS Pathogens

orcid.org/0000-0002-7699-2064

As you will see, all 3 reviewers were highly enthusiastic about this work and agreed on its importance to the field. However, some concerns were raised that require in some cases additional experimental work and clarification. In particular, the fate of the leader sequence of the cyt c1 protein requires further investigation, the question of whether atovaquone depolarizes the Toxoplasma mitochondrion is raised, and further controls are requested for loss of complex III signal upon knock-down of QCR proteins 8-12. It is also requested that the set of mitoribosomal proteins identified in the proteomic analysis is described in a stand-alone table, and a number of additional issues are raised that require attention.

Reviewer's Responses to Questions

**Part I - Summary**

Reviewer #1: In this manuscript, Maclean et al. used proteomic approaches (complexome profiling) to determine the protein composition of mitochondrial electron transport chain (mtETC) complexes and F1Fo-ATP synthase. The authors were able to confirm many subunit proteins that were previously identified in other proteomic studies. Importantly, they identified many new subunit proteins for the first time. In particular, in complexes II, IV and F1Fo-ATP synthase, the authors have identified 60 subunit proteins, of which 16 are novel proteins. In complex III, they have identified 4 new subunit proteins, of which 2 proteins are parasite-specific. They also experimentally characterized 4 new subunits of complex III to show their essentiality, mitochondrial localization, and complex III formation. Overall, using a combination of proteomic and genetic approaches, the authors have greatly expanded our knowledge of mtETC and F1Fo-ATP synthase in Toxoplasma gondii. This comprehensive new information is expected to aid structural and drug discovery studies to combat toxoplasmosis, malaria, and other infectious diseases caused by apicomplexan parasites. In addition to drug discovery, this manuscript will help us to understand the unique evolutionary biology of apicomplexan parasites. Overall, this work carried out by Maclean et al. represents one of comprehensive and beautiful studies in Toxoplasma gondii, which also has a broad impact on other apicomplexan parasites. The reviewer is highly enthusiastic about this work. However, the reviewer has one major concern and a few minor concerns, which need to be addressed before publication.

Highlights of this paper. Complex V is much bigger than previous prediction. The authors estimated its molecular weight to be ~1.860 MDa, much larger than previous predictions of 0.9 MDa or 1-1.2 MDa. Three new complex V subunits were also observed in the structure of complex V. One gene encodes two proteins that belong to two complexes, ATPG1 in complex V and cyt c1 in complex III. Also, this fusion architecture of two proteins is true for several apicomplexan parasites. The authors detected all mtDNA encoded proteins, cyt b, coxI and coxIII, which have not been experimentally detected before. This study identified likely, 6 novel subunits of complex IV. Complex II (~500 kDa) is much bigger than that of other organisms (~130 kDa). The authors identified 7 candidate novel complex II subunits, 4 of which are likely authentic complex II subunit proteins, albeit experimental data is yet available. They were able to identify 4 novel complex III proteins and provided sufficient biological verification data.

Reviewer #2: The authors have used BN-PAGE and mass spectrometry to fractionate and analyze the protein composition of mitochondrial respiratory complexes II-V. They identified novel subunits of all complexes and specifically tested the essentiality of 4 novel complex III sub-units. The mitochondrial ETC is a major drug target in Toxoplasma and Plasmodium parasites, but the conserved and divergent compositions of these complexes have not previously been studied in depth in Apicomplexan parasites. The present manuscript presents a valuable and impactful data-set that can form the basis for future tests to unravel functional properties of the parasite ETC, understand evolutionary and functional divergence between parasite and human complexes, and potentially identify novel drug targets that go beyond the current focus on cyt b. The study methodology is described in detail and appears to have been carefully performed.

Reviewer #3: Maclean and colleagues present their detailed study of the toxoplasma ETC using the well-regarded complexome technique. In establishing this in toxo they had to overcome significant hurdles regarding isolation of parasite mitochondria at relative purity, using the remnant host contaminants to calibrate their mass ranges in a really neat way. In providing an extensive overview of toxoplasma ETC subunit composition, they identify 16 new subunits of complexes II, III, IV and V. Outside of a comprehensive bioinformatics/phylogenetics analysis the authors go on to perform biochemical validation of four new Complex III subunits. This is significant since, as the authors discuss, CIII is the major drug target for apicomplexans, yet molecular composition in parasites is not fully clear. Importantly, the same four subunits are identified in another study currently available online as a preprint. As discussed by the authors, the competing study used a completely different approach making the two studies very complementary. The experiments are all performed to the highest standard and there are no experiments required to support the authors conclusions. Therefore, I only have minor comments, most of which can be addressed by changes in text.

**Part II – Major Issues: Key Experiments Required for Acceptance**

Reviewer #1: Major point

This study is carried out using the mitoribosomal tagged line (S35). Previously, this group has shown that mitoribosomal small subunit and large subunit were detected in BN-PAGE (Reference 34, Mol Micro 2019). Since proteomic approach of this study would detect every protein/complex from mitochondria, the authors need to show in separate tables or figures, the list of mitoribosomal proteins identified in this study.

Reviewer #2: 1. The authors’ identification that N-terminal cyt c1 peptides co-migrate with complex V is interesting, but the analysis and interpretation here is unclear and somewhat confusing. Cyt c1 is expected to have an N-terminal mitochondrial-targeting sequence that is cleaved upon import, based on studies in yeast (Pubmed: 2822392). The authors hint at possible processing but do not discuss this literature and instead imply that the N-terminal leader sequence and cyt c1 domains may be separate proteins expressed from the same gene. A simple model is that the N-terminal targeting sequence is retained after cleavage from cyt c1 upon mitochondrial import. This model would be analogous to the N-terminus of mammalian Rieske that is retained as part of complex III after cleavage (Pubmed: 8386158). Can the authors distinguish whether this N-terminal cyt c1 peptide is a stable subunit of complex V independent of interactions with complex III via the cyt c1 domain? For example, if the authors express an epitope-tagged copy of just the N-terminal ~150 amino acids of the cyt c1 protein, does it co-migrate with complex V by native PAGE and/or IP with complex V subunits? That experiment would substantially strengthen the very intriguing conclusion that this leader sequence associates with complex V independent of the cyt c1 domain.

2. The analysis and discussion of mitochondrial depolarization in figure 5B is confusing. (A) Is there prior published evidence that atovaquone treatment of T. gondii results in mitochondrial depolarization? The authors cite ref. 56 from Plasmodium as evidence that atovaquone depolarizes mitochondria, but follow-up studies in ref. 2 showed that ATV treatment alone does not depolarize the P. falciparum mitochondrion, due to the presence of a redundant, proguanil-sensitive proton-pumping pathway in blood-stage parasites thought to be the non-essential ATP synthase complex. T. gondii may differ in this regard from blood-stage Plasmodium, in which case inclusion of atovaquone in the figure 5B analysis would be helpful. (B) Interpretation of figure 5B would be strengthened if the authors can also include a histogram plot of the fluorescence shifts upon each QCR knockdown, along with representative epifluorescence images of individual parasites.

3. The authors present TOM40 detection in figure 6F as a negative control for specific loss of complex III signal upon knock-down of QCR proteins 8-12. The TOM40 signal is rather weak and the band position is not the same in all lanes, weakening the conclusion that QCR 8-12 knockdowns specifically destabilize complex III. Fig. 6 analysis would be strengthened by an additional negative control to rule out non-specific assembly defects due to general parasite malaise/death. For example, can the authors probe the blot in 6F for a different parasite protein (mitochondrial or cytosolic) that is not expected to be impacted by QCR knockdown, and/or treat parasites with a lethal inhibitor that targets something outside of the mitochondrion and show that ETC complex III stability is not impacted?

Reviewer #3: No major issues

**Part III – Minor Issues: Editorial and Data Presentation Modifications**

Reviewer #1: Minor points

1, Introduction-depiction of the Q cycle is inaccurate. Ubiquinol (QH2) binds to the Qo site and donates electrons via two paths, heam groups on cyt b and the Rieske protein. At the Qi site, ubiquinone (Q) not ubiquinol (QH2) accepts the electron passed by cyt b. It becomes a semi-quinone at the first cycle of electron transport and ubiquinol at the second cycle of electron transport.

2, Figure 2B, a typo of “totel”, a unnecessary “.” after the parenthesis. Supplementary Fig 8, “cat” repair should be cut repair. Page 13, Saccharomyces cerevisiae Cytochrome bc1, not b-c1. Page 32, Figure 5B legend, Valynomycin should be Valinomycin.

3, Figure 2B, it looks like mitochondria are distributed in every fraction of the Percoll gradient, from top, 18/23 interphase to 23/40 interphase. With equal loading (5 ug), it seems that mitochondria are not enriched too much, although contaminants are largely reduced. Also, apicoplast co-exists with mitochondria throughout the gradient. It has been known that apicoplast might form contact sites with mitochondria. Although apicoplast contamination is expected, the authors should still mention that in the text (Results), in addition to Discussion.

4, Figure S2A, it would be nice to add markers next to the gel to understand, roughly, the molecular weights of different complexes. S2B, this mass calibration shows complex V is smaller than 1000 kDa, isn’t it?

5, Figure 2C legend, this doi link points to the mitochondrial genome paper, not the mitochondrial proteome paper.

6, Supplementary data, which table is S2C, S2D?

7, Figure S3, it would be nice to specify the sequences of truncated VS untruncated peptides underneath the profiles to help readers to understand how host peptides screw the detection of parasite peptides.

8. Supplementary Fig. 5 legend, full names of Bb and Pf are missing.

9, Page 12. The doi link is for the mitochondrial genome paper, not the proteome paper. The same problem appeared again in Page 15.

10, Page 13. How did the author determine novel complex III subunits should be < 150 aa? It would be nice to show the sum of the molecular weights of 7 known subunits to indicate that estimation.

11, Page 32. Figure 5B, the numbers (% JC-1 positive cells) in the plots are invisible. Also, data of QCR11 should also be included in Figure 5B (albeit negative).

12, Page 33. Figure 6 legend, a “/” is missing between 9 and 11.

13, Page 16. Discussion of complex II. It is interesting that the authors disputed two previous predicated membrane anchor proteins of complex II in Plasmodium (PfSDH3, Pf3D7_0611100; PfSDH4, Pf3D7_1010300). It still remains unknown which subunit proteins are membrane anchor proteins of complex II since neither the previous study (2009, reference 61) nor this study performed biochemical/biological analysis. Also, could be large molecular weight a result of super-complex formation with complex II and other complexes?

14, Page 19, is it a Percoll gradient or sucrose gradient? If it is Percoll, it should be 18/23/40, not 15/23/40.

15, Page 20. It should be CytB, not COB.

16, Readers should be able to download the individual excel sheets of Supplementary data when the paper is published.

Reviewer #2: 4. The manuscript title seems narrowly focused on complex III, even though important insights were obtained for other complexes. The authors may wish to re-word to more accurately reflect the breadth of insights provided by their study, for example “Complexome profile of Toxoplasma gondii mitochondria identifies divergent subunits of respiratory chain complexes.”

5. The extensive supplemental information is very helpful but lacks figure and table labels on each page and is thus difficult to navigate.

6. The authors name TGGT1_214250 and TGGT1_207170 as QCR11 and QCR12, respectively. These names differ from the names for the same genes assigned in the Hayward study. To avoid confusion, the authors of both papers should presumably agree on a common name for these subunits.

7. The authors do not assess the degree of knockdown obtained +ATc for QCR proteins 8/9 and 11/12 in figures 5 and 6. Western blot analysis may not be possible if these proteins were not epitope-tagged, but the authors can at least assess mRNA reduction by RT-qPCR.

Reviewer #3: -Panel S2B should be changed to C, and C, D to B, C to reflect order discussed in text.

-Supplemental Tables (the large proteomics/informatics ones) were presented as excel files saved in the pdf format. I acknowledge that this could be as a result of the journal upload system and was not the intention of the authors. If this was intended please replace by actual excel (or csv) sheets as presentation in pdf format is not really useful to the reader since the formatting is lost (I found it very frustrating and just gave up on them, so this is a caveat in my review!).

-The authors often compare their complex mass estimates to other publications that used different detergents, most of which alter the migration of complexes on BN-PAGE either due to dissociation of subunits or possible assembly into supercomplexes (the latter is nicely illustrated in Fig 4F). While this is mentioned in the discussion, given the authors discovery of parasite supercomplexes that can be resolved using digitonin, and the fact that some of said other studies used digitonin or TX-100 in their size calculations (vs DDM for this study) I think it is fair to raise this caveat at the initial point of mention in the results.

-A similar issue on mass estimates arises in the CII and III sections where the authors quote the digitonin resolved sizes of 130 kDa and 500 kDa (Schaegger 2000 EMBO J). Perhaps it is safer when referring to mammalian complexes to use the absolute calculations from Hirst, Kunji, Walker 2019 Science?

-The CIII2+IV SC mass used in the authors calibration appears to be CIII2+IV2 (Table S4) though this isn’t reflected in the figure. Is this SC assembly stable in DDM? Please clarify and if no, revise calculations appropriately.

-Are any of the new CIV (or III) subunits the homolog of NDUFA4, which is known to be present in P. falciparum (Balsa 2012 Cell Metab)?

-The text describes the SDHA subunit profile being split into two peaks due to some peptides being identical to the host SDHA peptides, similar as for ATPa/b subunits. In this case however, once considering only the parasite specific peptides the peak still co-migrated with the host complex (slice 35 vs slice 18 for the parasite). Does this mean the toxoplasma SDHA subunit isn’t associated with the other Complex II subunits (and possibly not a CII subunit at all)?

-There appears to be a typo “In addition to complex II, we observed evidence for the following quinone reductases (Figure 1B): two single-subunit type II NADH dehydrogenases (NDH2)…” on p12 since Figure 1B has nothing to do with this text.

-Putative CIII subunit (later ruled out) TGGT1_297160 is not listed in Table 2 though discussed extensively in text. Since the only other place to refer to it is the large Table S1 (which is rendered in an unreadable format) I think including it here plus indicating that it was excluded in the legend will be less confusing to readers.

-It would be helpful to include the new protein names (next to the original TGGT1_xxx) in parentheses on Fig 2B

-Regarding QCR7 and the conclusion that the low relative levels indicates that it may be expressed at a lower level. Can you exclude the HA tag impacting assembly into the complex and therefore leading to its turnover?

PLOS authors have the option to publish the peer review history of their article (what does this mean?). If published, this will include your full peer review and any attached files.

Reviewer #1: **Yes: **Hangjun Ke

Reviewer #2: No

Reviewer #3: No
---

## [Decision Letter · Decision Letter 1]

16 Dec 2020

Dear Lilach,

Thank you very much for submitting your manuscript "Complexome profile of Toxoplasma gondii mitochondria identifies divergent subunits of respiratory chain complexes including new subunits of cytochrome bc1 complex" for consideration at PLOS Pathogens. As with all papers reviewed by the journal, your manuscript was reviewed by members of the editorial board and by several independent reviewers. The reviewers appreciated the attention to an important topic. Based on the reviews, we are likely to accept this manuscript for publication, providing that you modify the manuscript according to the review recommendations.

As you will see, whilst Reviewer #2 is satisfied that the revised manuscript is now worthy of consideration for publication, the other two reviewers would like some minor concerns and clarifications to be addressed. Specifically, Reviewer #1 would like you to clarify the rigour with which the mass spectrometry data have been determined, with particular attention to those proteins identified on the basis of a single peptide and the number of repicate experiments used to assign identification. The reviewer also requests some modifications to the supplementary tables providing the mass spectrometric data, intended to enhance clarity. Please make these data fully available through ToxoDB. Reviewer #3 requests some minor changes to the nomenclature used in Figure 2D, and also that references to unpublished data are presented in a manner consistent with journal policy. We trust that these amendments will not be unduly onerous.

Sincerely,

Michael J Blackman

Associate Editor

PLOS Pathogens

Vern Carruthers

Section Editor

PLOS Pathogens

Kasturi Haldar

Editor-in-Chief

PLOS Pathogens

orcid.org/0000-0001-5065-158X

Michael Malim

Editor-in-Chief

PLOS Pathogens

orcid.org/0000-0002-7699-2064

As you will see, whilst Reviewer #2 is satisfied that the revised manuscript is now worthy of consideration for publication, the other two reviewers would like some minor concerns and clarifications to be addressed. Specifically, Reviewer #1 would like you to clarify the rigour with which the mass spectrometry data have been determined, with particular attention to those proteins identified on the basis of a single peptide and the number of repicate experiments used to assign identification. The reviewer also requests some modifications to the supplementary tables providing the mass spectrometric data, intended to enhance clarity. Please make these data fully available through ToxoDB. Reviewer #3 requests some minor changes to the nomenclature used in Figure 2D, and also that references to unpublished data are presented in a manner consistent with journal policy. We trust that these amendments will not be unduly onerous.

Reviewer Comments (if any, and for reference):

Reviewer's Responses to Questions

**Part I - Summary**

Reviewer #1: The revised version of this manuscript by Maclean et al. has improved a lot in data clarity and quality. The authors have nicely responded to reviewers' comments point-by-point. The reviewer appreciates the authors' hard work.

Reviewer #2: The authors have revised their prior submission to strongly address reviewer critiques and suggestions. I assume that the mass spectrometry data generated in this study will be made available via the ToxoDB interface, including individual peptides detected for each protein. Such information will be a valuable resource for the community.

Reviewer #3: The authors have made a number of important changes to yield a high quality manuscript of general interest to the community. In doing so they addressed most of my key concerns, however there are two very minor issues indicated below that should be resolved prior to publication.

**Part II – Major Issues: Key Experiments Required for Acceptance**

Reviewer #1: After carefully reviewing the mass spectrometry data, the reviewer realized one issue the authors might have missed to address. Among 842 proteins identified (Table S2), 381 had only one peptide. When the detection limit is low, how to differentiate a signal from a noise? Have the authors repeated this complexome profiling several times?

Reviewer #2: The authors have strongly addressed the prior critiques.

Reviewer #3: (No Response)

**Part III – Minor Issues: Editorial and Data Presentation Modifications**

Reviewer #1: After carefully reviewing the revised paper, the reviewer still has several minor issues.

1, The complexome profiling data identified 842 Toxoplasma proteins in total, listed in table S2. The authors mentioned 264 of those are mitochondrial. Could the authors list these 264 proteins in a sperate excel table? Also, please list gene ID, protein annotation, peptide count, etc., in separate excel columns. As it now, column A of S2 is hard to read with so much information in one cell (TGGT1:KE387283:30672:39285:-1 gene:TGGT1_243490:EPR56544 |putative BCS1 family isoform 9). Also, importantly, could the authors make the new table even more convenient to readers by designating some of the hypothetical proteins (in database) to known functions? At least, the authors have validated some unknown proteins in this study.

2, Please list correct gene IDs of orthologs in other parasites in Table S7. The top hit IDs in table S7 are not ready for database searches, at least in a convenient manner.

3, Legend of Table S1 lacks how the authors calculated abundance. How did the authors convert peptide counts to percentage of abundance? This math is critical to all analyses of this paper.

Reviewer #2: (No Response)

Reviewer #3: - In respect to the SDHA homolog being not associated with CII in complexome: I thank the authors for sharing their IP data confirming association of SDHA with SDHB destined for another manuscript. I'm undecided if it is sufficient to state "IP experiments of complex II using either SDHB as bait consistently recover SDHA (Maclean et al in preparation).". The data availability of PLoS Pathogens (https://journals.plos.org/plospathogens/s/submission-guidelines) states that 'data not shown' is not an acceptable citation. If the editor agrees and if the other manuscript has been published or released as a preprint please replace with correct citation, otherwise remove. If removed I suggest the authors mention the caveat and suggest the IP as a method to confirm SDHA association with the complex in support of the other evidence cited in lines 526-548.

- In respect to revised figure 2D I thank the authors for inclusion of the new names for CIII subunits. The revised discussed also renames a number of other subunits (e.g. TgCox4) so these should also be included in the figure for consistency.

PLOS authors have the option to publish the peer review history of their article (what does this mean?). If published, this will include your full peer review and any attached files.

Reviewer #1: No

Reviewer #2: No

Reviewer #3: No
---

## [Editor Report · Decision Letter 2]

11 Jan 2021

Dear Lilach,

We are pleased to inform you that your manuscript 'Complexome profile of Toxoplasma gondii mitochondria identifies divergent subunits of respiratory chain complexes including new subunits of cytochrome bc1 complex' has been provisionally accepted for publication in PLOS Pathogens.

Best regards,

Michael J Blackman

Associate Editor

PLOS Pathogens

Vern Carruthers

Section Editor

PLOS Pathogens

Kasturi Haldar

Editor-in-Chief

PLOS Pathogens

orcid.org/0000-0001-5065-158X

Michael Malim

Editor-in-Chief

PLOS Pathogens

orcid.org/0000-0002-7699-2064
---

## [Editor Report · Acceptance letter]

24 Feb 2021

Dear Dr. Sheiner,

We are delighted to inform you that your manuscript, "Complexome profile of Toxoplasma gondii mitochondria identifies divergent subunits of respiratory chain complexes including new subunits of cytochrome bc1 complex ," has been formally accepted for publication in PLOS Pathogens.

Best regards,

Kasturi Haldar

Editor-in-Chief

PLOS Pathogens

orcid.org/0000-0001-5065-158X

Michael Malim

Editor-in-Chief

PLOS Pathogens

orcid.org/0000-0002-7699-2064